# Turing Completeness of Bounded-Precision Recurrent Neural Networks

**Stephen Chung**[*]
Department of Computer Science
University of Massachusetts Amherst
Amherst, MA 01003
`minghaychung@umass.edu`

**Hava Siegelmann**[*]
Department of Computer Science
University of Massachusetts Amherst
Amherst, MA 01003
`hava@umass.edu`

## Abstract

Previous works have proved that recurrent neural networks (RNNs) are Turing-complete. However, in the proofs, the RNNs allow for neurons with unbounded precision, which is neither practical in implementation nor biologically plausible. To remove this assumption, we propose a dynamically growing memory module made of neurons of fixed precision. The memory module dynamically recruits new neurons when more memories are needed, and releases them when memories become irrelevant. We prove that a 54-neuron bounded-precision RNN with growing memory modules can simulate a Universal Turing Machine, with time complexity linear in the simulated machine's time and independent of the memory size. The result is extendable to various other stack-augmented RNNs. Furthermore, we analyze the Turing completeness of both unbounded-precision and bounded-precision RNNs, revisiting and extending the theoretical foundations of RNNs.

## 1 Introduction

Symbolic (such as Turing Machines) and sub-symbolic processing (such as adaptive neural networks) are two competing methods of representing and processing information, each with its own advantages. An ultimate way to combine symbolic and sub-symbolic capabilities is by enabling the running of algorithms on a neural substrate, which means a neural network that can simulate a Universal Turing Machine (UTM). Previous works [1, 2, 3] have shown that this is possible – there exists a recurrent neural network (RNN) that can simulate a UTM. These proofs assumed a couple of neurons with unbounded precision that equals the number of symbols used in the Turing tape. Here we provide an alternative simulation of a UTM by RNNs with bounded-precision neurons only.

The general idea works as follows. The Turing Machine's tape is stored in a *growing memory module*, which is a stack of neurons with pushing and popping operations controlled by neurons in the RNN. The size of the growing memory module is determined by the usage of the Turing tape - it dynamically recruits new neurons when more memories are needed and releases them when memories become irrelevant. The neurons in the stack, except for the top neuron, are not regularly updated (and hence can be referred to as *passive*), saving computational cost for memories that are not in the focus of the computing and do not require change. Using growing memory modules, a 54-neuron bounded-precision RNN is constructed that can simulate any Turing Machine.

Our proposed growing memory modules are inspired by biological memory systems. The process of dynamically recruiting new neurons when more memories are necessary is also observed in biological memory systems. Neurogenesis is the process by which new neurons are produced in the central nervous system; it is most active during early development, but continues through life.

---

[*]Both authors contributed equally.

35th Conference on Neural Information Processing Systems (NeurIPS 2021).

In adult vertebrates, neurogenesis is known to occur in the dentate gyrus (DG) of the hippocampal formation [4] and the subventricular zone (SVZ) of the lateral ventricles [5]. Since DG is well-known in neuroscience for its role in pattern separation for memory encoding [6, 7], this suggests that biological memory systems also dynamically recruit new neurons. The rate of neurogenesis in adult mice has been shown to be higher if they are exposed to a wider variety of experiences [8]. This further suggests a role for self-regulated neurogenesis in scaling up the number of new memory that can be encoded and stored during one's lifetime without catastrophic forgetting of previously consolidated memory. Besides the mechanism of recruiting new neurons, the process of storing neurons in growing memory modules also shares some similarities with biological memory consolidation, a process by which short-term memory is transformed into long-term memory [9, 10]. Compared to short-term memory, long-term memory is more long-lasting and robust to interference. This is similar to the neurons stored in growing memory modules - the values of these neurons (except the top neuron in the stack) remain unchanged and cannot be interfered by the RNN, providing a mechanism to store information stably.

Growing memory modules share similarities with other stack-augmented RNNs [11, 12, 13, 14, 15]. In neural stacks [11], the RNN outputs two continuous scalars that control the strength of pushing and popping operations, and the stack is made differentiable by adding a strength vector. In stack RNNs [12] and DiffStk-RNN [15], the RNN outputs the probability vector corresponding to pushing, popping, and no operation. In NNPDA [13], the RNN outputs a continuous scalar that controls the pushing and popping operations, with the minimum value corresponding to popping and the maximum value corresponding to pushing. NSPDA [14] uses a discrete-valued action neurons to control pushing and popping operations. In contrast to these models, growing memory modules have a simple design and do not need to be differentiable. However, it can be easily shown that growing memory modules can be simulated by these stack-augmented RNNs in linear time, and thus growing memory modules can be considered a generic type of stack-augmented RNNs. Therefore, our proof on the Turing completeness of an RNN with growing memory modules can be extended to stack-augmented RNNs in general, establishing their theoretical motivation.

A Turing-complete RNN that is fully differentiable was introduced in 1996 [16]; this feature is a prerequisite to have the network trainable by gradient descent. It was followed by the Neural Turing Machine (NTM) [17] and its improved version, the differentiable neural computer [18], which are both differentiable and trainable RNNs equipped with memory banks. Though inspired by Turing Machines, bounded-precision NTMs and differentiable neural computers are not Turing-complete due to the fixed-sized memory bank, but they can simulate space-bounded Turing Machines (see Section 5).

Simulation of a Turing Machine by an RNN with growing memory modules represents a practical and biologically inspired way to combine symbolic and sub-symbolic capabilities. All neurons in the RNN and growing memory modules have fixed precision. While the size of growing memory modules is linear in the number of symbols used in the Turing tape, the number of neurons in the RNN is still constant. Moreover, the neurons in growing memory modules (except the top neuron in the stack) are passive at most times. As a result, the time complexity of the simulation is linear in the simulated machine's time and independent of the memory size. By showing how to simulate a Turing Machine with a bounded-precision RNN and thereby constructing a bounded-precision RNN that can run any algorithms, our paper proposes a practical method that combines symbolic and sub-symbolic capabilities.

The remainder of the paper is structured as follows. Section 2 describes the preliminary of the paper, including the definition of Turing Machines and RNNs. Section 3 revisits and extends theories relating to simulating a Turing Machine with unbounded-precision RNNs and shows the existence of a 40-neuron unbounded-precision RNN that is Turing-complete. Section 4 presents the growing memory modules and proves the existence of a 54-neuron bounded-precision RNN with two growing memory modules that is Turing-complete. Section 5 relates the number of neurons and the precision of RNNs when simulating Turing Machines. Section 6 concludes the paper.

## 2   Background and Notation

A Turing Machine is a 7-tuple $\mathcal{M} = (Q, \Sigma, \Gamma, \delta, q_0, \sharp, F)$, where $Q$ is a finite set of states, $\Sigma$ is a finite set of input symbols, $\Gamma$ is a finite set of tape symbols (note that $\Sigma \subset \Gamma$), $\delta : Q \times \Gamma \to Q \times \Gamma \times \{L, R\}$

is the machine's transition rule, $q_0 \in Q$ is the initial starting state, $\sharp$ is the blank symbol (note that $\sharp \in \Gamma$ but $\sharp \notin \Sigma$), and $F \subset Q$ is the set of final state. We only consider deterministic Turing Machines in this paper.

The *instantaneous description* (or the configuration) of a Turing Machine is typically defined as a tuple of state, tape, and the location of the read/write head. However, we use a slightly different definition in the paper. We define the *left-tape symbols* (or the left tape in short), denoted as $s_L$, to be the string of symbols starting at the symbol under the read/write head and extending to the left. We define the *right-tape symbols* (or the right tape in short), denoted as $s_R$, to be the string of symbols starting at the symbol to the right of the read/write head and extending to the right. The first symbol in both strings is the closest to the read/write head in both $s_L$ and $s_R$ (that is, the left tape is reversed in the representation) and the blank symbols at the two ends are omitted. Therefore, the length of $s_L$ and $s_R$ combined equals to the number of symbols used in the Turing tape in each step (that is, unbounded but not infinite). Since $s_L$ and $s_R$ encode both the tape and the location of the read/write head, we define the instantaneous description of a Turing Machine as a 3-tuple $(q, s_L, s_R) \in (Q, \Gamma^*, \Gamma^*)$, where $q$ denotes the state, $s_L$ denotes the left-tape symbols, and $s_R$ denotes the right-tape symbols. Though the two definitions are equivalent, this definition allows easy encoding of tape symbols into neurons' values. The set of all possible instantaneous description is denoted as $\mathcal{X} := (Q, \Gamma^*, \Gamma^*)$.

In each step of a Turing Machine, the symbol under the read/write head is read and, together with the state, determines the symbol to be written under the read/write head, the direction of moving the tape, and the next state. To be precise, the *complete dynamic map* of $\mathcal{M}$, denoted as $\mathcal{P}_\mathcal{M} : \mathcal{X} \to \mathcal{X}$, is defined as follows: 1. Let $x = (q, s_L, s_R)$ be the input configuration, $s_{L,(1)}$ denote the first symbol in $s_L$ and $s_{R,(1)}$ denote the first symbol in $s_R$. The transition is defined by the 3-tuple $(q', y, d) = \delta(q, s_{L,(1)})$; 2. Replace the state of the machine $q$ with $q'$ and the first symbol in $s_L$ by $y$; 3. Move the symbol in $s_{L,(1)}$ to become the new $s_{R,(1)}$ if $d = L$, and move $s_{R,(1)}$ to become the new $s_{L,(1)}$ if $d = R$ (if there are no symbols left in $s_L$ or $s_R$ for moving, append a blank symbol $\sharp$ to it before moving). Denote the left-tape symbols and the right-tape symbols after 2. and 3. by $s'_L$ and $s'_R$ respectively. Then $\mathcal{P}_\mathcal{M}(x) = (q', s'_L, s'_R)$ represents one transition of the Turing Machine $\mathcal{M}$ from one configuration to the next. The *partial input-output function* of $\mathcal{M}$, denoted as $\mathcal{P}^*_\mathcal{M} : \mathcal{X} \to \mathcal{X}$, is defined by applying $\mathcal{P}_\mathcal{M}$ repeatedly until $q \in F$, and is undefined if it is not possible to have $q \in F$ by applying $\mathcal{P}_\mathcal{M}$ repeatedly.

A recurrent neural network (RNN) is a neural network consisting of $n$ neurons. The value of neuron $i$ at time $t \in \{1, 2, ...\}$, denoted as $x_i(t) \in \mathbb{Q}$ ($\mathbb{Q}$ is the set of rational numbers), is computed by an affine transformation of the values of neurons in the previous state followed by an activation function $\sigma$, i.e. $x_i(t) = \sigma(\sum_{j=1}^n w_{ij} x_j(t-1) + b_i)$, where $w_{ij}$ are the weights and $b_i$ is the bias; or in vector form:

$$\mathbf{x}(t) = \sigma(W\mathbf{x}(t-1) + \mathbf{b}), \tag{1}$$

where $\mathbf{x}(t) \in \mathbb{Q}^n$, $W \in \mathbb{R}^{n \times n}$ and $\mathbf{b} \in \mathbb{R}^n$. This defines a mapping $\mathcal{T}_{W,\mathbf{b}} : \mathbb{Q}^n \to \mathbb{Q}^n$ which characterizes an RNN. For simplicity, we consider the saturated-linear function in this paper; that is:

$$\sigma(x) := \begin{cases} 0 & \text{if } x < 0, \\ x & \text{if } 0 \le x \le 1, \\ 1 & \text{if } x > 1. \end{cases} \tag{2}$$

Thus, $\mathbf{x}(t) \in (\mathbb{Q} \cap [0, 1])^n$ for all $t > 0$.

We say that a neuron $x_i(t)$ has precision $p$ in base $b$ if for all $t > 0$, $x_i(t)$ can be expressed as $\sum_{l=1}^p \frac{a_{(l)}(t)}{\prod_{j=1}^l c_{(j)}(t)}$ for some strings $a(t) \in \{0, 1, ..., b\}^p$ and $c(t) \in \{1, ..., b\}^p$.

For a string $a$, we use $a_{(i)}$ to denote the $i^{\text{th}}$ symbol in $a$ and $a_{(i:j)}$ to denote the string $a_{(i)}a_{(i+1)}...a_{(j)}$. For a function $f$ that maps from a set $\mathbb{Y}$ to a subset of $\mathbb{Y}$, we use $f^n$ to denote the $n^{\text{th}}$ iterate of $f$ where $1 \le n < \infty$. For any two vectors $\mathbf{x} \in \mathbb{R}^m, \mathbf{y} \in \mathbb{R}^n$, we use $\mathbf{x} \oplus \mathbf{y} \in \mathbb{R}^{m+n}$ to denote the concatenation of the two vectors. We use $A^*$ to denote all possible strings formed by elements from set $A$.

The notation is summarized in the table found in Appendix D.

## 3  Turing Completeness of Unbounded-Precision RNNs

To simulate a Turing Machine $\mathcal{M}$ by an RNN, we first consider how to encode the instantaneous description $(q, s_L, s_R) \in \mathcal{X}$ by a vector of rational numbers. For the state $q \in Q$, we encode it with $\lceil \log_2 |Q| \rceil$ binary values, denoted as $\rho^{(q)} : Q \to \{0, 1\}^{\lceil \log_2 |Q| \rceil}$, with each possible combination of binary values representing a specific state.

*Example.* Let $Q = \{1, 2, 3, 4, 5, 6\}$. We can encode it by $\lceil \log_2 |Q| \rceil = 3$ binary values, with $\rho^{(q)}(1) = [0, 0, 0], \rho^{(q)}(2) = [0, 0, 1], \rho^{(q)}(3) = [0, 1, 0], \rho^{(q)}(4) = [0, 1, 1], \rho^{(q)}(5) = [1, 0, 0]$, and $\rho^{(q)}(6) = [1, 0, 1]$.

For the left tape $s_L \in \Gamma^*$ and the right tape $s_R \in \Gamma^*$, we use fractal encoding in two rational numbers, as recommended in [1, 2]. The fractal encoding, bearing similarity to Cantor sets, enables fast manipulation of the top symbols.

Without loss of generality, assume that the tape symbols $\Gamma$ are encoded into numbers $\{1, 3, 5, ..., 2|\Gamma| - 1\}$ and that the blank symbol $\sharp$ is encoded by 1. Then, define the fractal encoding $\rho^{(s)} : \Gamma^* \to \mathbb{Q}$ by:

$$\rho^{(s)}(y) := \left( \sum_{i=1}^{|y|} \frac{y_{(i)}}{(2|\Gamma|)^i} \right) + \frac{1}{(2|\Gamma|)^{|y|} \cdot (2|\Gamma| - 1)}. \tag{3}$$

*Example.* Let $\Gamma = \{1, 3, 5, 7\}$, $s_L = (3, 5, 7, 3, 5)$, and $\sharp = 1$. Then $\rho^{(s)}(s_L) = \frac{3}{8} + \frac{5}{8^2} + \frac{7}{8^3} + \frac{3}{8^4} + \frac{5}{8^5} + \frac{1}{8^6} + \frac{1}{8^7} + \frac{1}{8^8} + ... = \frac{3}{8} + \frac{5}{8^2} + \frac{7}{8^3} + \frac{3}{8^4} + \frac{5}{8^5} + \frac{1}{8^5 \cdot 7}$. The last term in (3) represents the infinite blank symbols of a tape.

This encoding requires the tape symbol neurons to have the same precision as the size of the active (non-blank) part of the tape in base $2|\Gamma|$. As the tape in a Turing Machine has an unbounded size, it means that we require neurons with unbounded precision. This is different from infinite precision but still not applicable. Hence, we will discuss how to remove this unbounded-precision assumption in Section 4.

Finally, we encode the value of the top symbol in each tape neuron, $s_{L,(1)}$ and $s_{R,(1)}$, by a binary tuple $\rho^{(r)} : \Gamma \to \{0, 1\}^{|\Gamma| - 1}$:

$$\rho_i^{(r)}(y) := 1\{y > 2i\} \qquad (1 \le i \le |\Gamma| - 1). \tag{4}$$

That is, the $i$ coordinate of $\rho^{(r)}(y)$ is 1 if and only if the symbol $y$ has a value larger than $2i$, and is 0 otherwise.

*Example.* Let $\Gamma = \{1, 3, 5, 7\}$. Then $\rho^{(r)}(1) = [0, 0, 0]$, $\rho^{(r)}(3) = [1, 0, 0]$, $\rho^{(r)}(5) = [1, 1, 0]$, and $\rho^{(r)}(7) = [1, 1, 1]$.

Combining the above discussion, we define the encoding function of configurations $\rho : \mathcal{X} \to \mathbb{Q}^{2|\Gamma| + \lceil \log_2 |Q| \rceil + |Q||\Gamma| + 5}$ by:

$$\rho(q, s_L, s_R) = \rho^{(q)}(q) \oplus \rho^{(s)}(s_L) \oplus \rho^{(s)}(s_R) \oplus \rho^{(r)}(s_{L,(1)}) \oplus \rho^{(r)}(s_{R,(1)}) \oplus \mathbf{0}, \tag{5}$$

where $\mathbf{0}$ is a zero vector of size $|Q||\Gamma| + 5$. Let $\rho^{-1} : \rho(\mathcal{X}) \to \mathcal{X}$ be the inverse function such that $\rho^{-1}(\rho(x)) = x$ for all $x \in \mathcal{X}$ (note that $\rho$ is injective, i.e. $\rho(x) = \rho(x')$ implies $x = x'$); we call $\rho^{-1}$ the decoder function.

It should be noted that $\rho^{(q)}(q) \oplus \rho^{(s)}(s_L) \oplus \rho^{(s)}(s_R)$ is sufficient to decode the instantaneous description. We include $\rho^{(r)}(s_{L,(1)}) \oplus \rho^{(r)}(s_{R,(1)}) \oplus \mathbf{0}$ only because it facilitates the full simulation on the RNN. This completes the construction of the encoding function.

Given an instantaneous description $x \in \mathcal{X}$, we initialize the neurons in an RNN with values $\rho(x)$. Then, it is possible to construct the parameters of RNN such that the update given by the RNN on these neurons is the same as the update given by the Turing Machine:

**Theorem 1.** *Given a Turing Machine $\mathcal{M}$, there exists an injective function $\rho : \mathcal{X} \to \mathbb{Q}^N$ and an $n$-neuron unbounded-precision RNN $\mathcal{T}_{W,\mathbf{b}} : \mathbb{Q}^n \to \mathbb{Q}^n$, where $n = 2|\Gamma| + \lceil \log_2 |Q| \rceil + |Q||\Gamma| + 5$, such that for all instantaneous descriptions $x \in \mathcal{X}$,*

$$\rho^{-1}(\mathcal{T}_{W,\mathbf{b}}^3(\rho(x))) = \mathcal{P}_{\mathcal{M}}(x). \tag{6}$$

*Proof.* A sketch of the proof is as follows. Neurons in the RNNs are grouped by their function. Tape neurons, initialized with $\rho^{(s)}(s_L)$ and $\rho^{(s)}(s_R)$, encode the tape in fractal encoding. Readout neurons, initialized with $\rho^{(r)}(s_{L,(1)})$ and $\rho^{(r)}(s_{R,(1)})$, encode the first symbol in the left and the right tape. State neurons, initialized with $\rho^{(q)}(q)$, encode the state. We need to update the values of these neurons to simulate one step of the Turing Machine. Three steps (or *stages*) of RNN are required to simulate one step of a Turing Machine. In the first stage, entry neurons, initialized with 0, compute the combination of the state and the symbol under the head. Since this combination fully determines the next transition, we use it to update the state neurons and the temporary tape neurons during stage two. Temporary tape neurons serve as a buffer for tape neurons when shifting the tape to the left or right. In stage three, we move the values from the temporary tape neurons to tape neurons to complete the update. The detailed proof can be found in Appendix A. □

In other words, to simulate one step of a Turing Machine $\mathcal{M}$, we can first encode the instantaneous description $x$ by the encoder function $\rho$, apply the RNN three times $\mathcal{T}_{W,\mathbf{b}}^3$, and decode the values back by $\rho^{-1}$ to obtain $\mathcal{P}_{\mathcal{M}}(x)$, the instantaneous description after one step of the Turing Machine. Or equivalently, for any Turing Machine $\mathcal{M}$, there exists an RNN such that every three steps of the RNN yield the same result as one step of the Turing Machine.

By applying Theorem 1 repeatedly, we simulate a Turing Machine with an RNN in a linear time. To be specific, the *partial input-output function* of an RNN, denoted as $\mathcal{T}_{W,\mathbf{b}}^* : \mathbb{Q}^N \to \mathbb{Q}^N$, is defined by applying $\mathcal{T}_{W,\mathbf{b}}^3$ repeatedly until $q \in F$ (where $q$ is the state that the RNN simulates), and is undefined if it is not possible to have $q \in F$ by applying $\mathcal{T}_{W,\mathbf{b}}^3$ repeatedly. Based on this definition and Theorem 1, it follows that:

**Corollary 1.1.** *Given a Turing Machine $\mathcal{M}$, there exists an injective function $\rho : \mathcal{X} \to \mathbb{Q}^n$ and an $n$-neuron unbounded-precision RNN $\mathcal{T}_{W,\mathbf{b}} : \mathbb{Q}^n \to \mathbb{Q}^n$, where $n = 2|\Gamma| + \lceil \log_2 |Q| \rceil + |Q||\Gamma| + 5$, such that for all instantaneous descriptions $x \in \mathcal{X}$, the following holds: If $\mathcal{P}_{\mathcal{M}}^*(x)$ is defined, then*

$$\rho^{-1}(\mathcal{T}_{W,\mathbf{b}}^*(\rho(x))) = \mathcal{P}_{\mathcal{M}}^*(x), \tag{7}$$

*and if $\mathcal{P}_{\mathcal{M}}^*(x)$ is not defined, then $\mathcal{T}_{W,\mathbf{b}}^*(\rho(x))$ is also not defined. If $\mathcal{P}_{\mathcal{M}}^*(x)$ is defined and computed in $T$ steps by $\mathcal{M}$, then $\mathcal{T}_{W,\mathbf{b}}^*(\rho(x))$ is computed in $3T$ steps by the RNN.*

Corollary 1.1 shares similarities with Theorem 1 in [1]. However, our theorem states that $3T$, instead of $4T$, is sufficient to simulate a Turing Machine. We also give the relationship between the number of neurons required by the RNN and the size of $Q$ and $\Gamma$ in the Turing Machine.

A small UTM with 6 states and 4 symbols, denoted by $U_{6,4}$, was proposed [19] and can simulate any Turing Machine in time $\mathcal{O}(T^6)$, where $T$ is the number of steps required by the Turing Machine (the one to be simulated) to compute the result. As $U_{6,4}$ is also a Turing Machine, we apply Corollary 1.1 to simulate $U_{6,4}$, leading to a Turing-complete RNN. Plugging in $|Q| = 6$ and $|\Gamma| = 4$, we obtain the following result:

**Corollary 1.2.** *There exists a 40-neuron unbounded-precision RNN that can simulate any Turing Machine in $\mathcal{O}(T^6)$, where $T$ is the number of steps required for the Turing Machine to compute the result.*

It should be noted that [1] focused on capabilities and proved a Turing-complete RNN with 1058 neurons; [20] proposed a Turing-complete RNN with 52 neurons. Here we provide a plug-and-play formula to simulate any Turing Machine.

## 4  Turing Completeness of Bounded-Precision RNNs with Growing Memory

In the following, we consider how to remove the assumption of unbounded neural precision (Section 3) without reducing computational capacity. If we assume all neurons to have precision bounded by $p$ in base $2|\Gamma|$, then the tape can be encoded by $\lceil |s_j|/p \rceil$ neurons using fractal encoding, where $j \in \{L, R\}$. We do this by encoding every $p$ symbols of the tape into a single neuron. Since most of these neurons do not require updates, just like symbols far from the read/write head in a Turing tape, we propose to store them in a separate *growing memory module* organized in a stack-like manner:

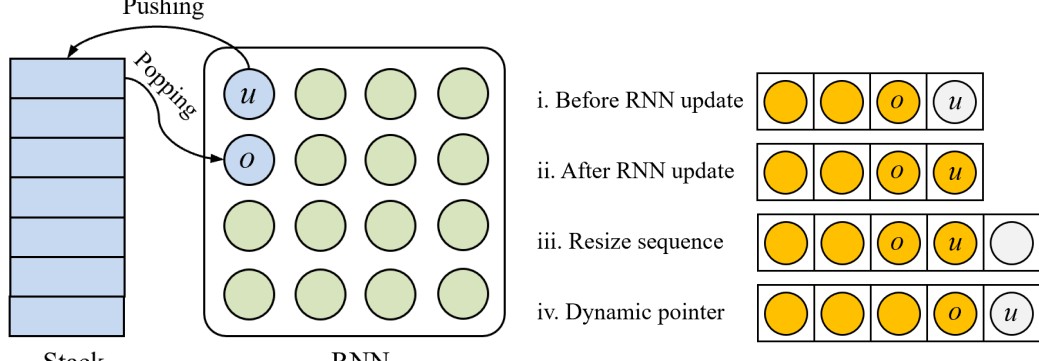

(a) An RNN with a growing memory module          (b) Pushing with dynamic pointer

Figure 1: (a) An RNN with a growing memory module. The RNN controls the pushing and popping of the stack by two neurons $u(t)$ and $o(t)$. (b) Illustration of pushing in growing memory module as a dynamic pointer. Orange (dark) and gray (light) circles represent non-zero neurons and zero neurons respectively. i. Before RNN update, $u(t-1) = 0$. ii. RNN set $u(t) = c$ where $c > 0$ is the value to be pushed. iii. One zero neuron is appended to the sequence since there are no zero neurons left. iv. Both $u$ and $o$ point to different neurons in the sequence, ready to be read by other neurons in the next update of RNN.

**Definition 1.** *A growing memory module is a stack of non-zero neurons with push and pop operations controlled by two neurons in an RNN, denoted as push neuron $u(t)$ and pop neuron $o(t)$, in the following way: for every step $t$ after the RNN finished updating the values of neurons, (i) if $u(t) > 0$, then a new neuron with the value $u(t)$ is pushed to the stack and $u(t)$ is set to 0; (ii) if $o(t) = 0$ and the stack is not empty, then the top neuron is popped from the stack and $o(t)$ is set to the value of the top neuron in the updated stack; (iii) if $o(t) = 0$ and the stack is empty, then $o(t)$ is set to a default value $c$.*

An RNN with a growing memory module is a mapping $\mathcal{T}_{W,\mathbf{b}} : (\mathbb{Q}^N, \mathbb{Q}^*) \to (\mathbb{Q}^N, \mathbb{Q}^*)$ where the first element of the tuple corresponds to values of neurons in the RNN and the second element of the tuple corresponds to the stack of the growing memory module. We can equip an RNN with multiple growing memory modules, with each module having its own push and pop neurons controlled by the RNN. An RNN with two growing memory modules will be defined by a mapping $\mathcal{T}_{W,\mathbf{b}} : (\mathbb{Q}^N, \mathbb{Q}^*, \mathbb{Q}^*) \to (\mathbb{Q}^N, \mathbb{Q}^*, \mathbb{Q}^*)$.

We can view the growing memory module as a way of dynamically pointing to a sequence of non-zero neurons appended by one zero neuron. $o(t)$ can be viewed as the pointer for the last non-zero neuron in the sequence, and $u(t)$ can be viewed as the pointer for the zero neuron at the beginning of the sequence; see Figure 1.

With two growing memory modules (one for the left tape and one for the right tape), we can construct an RNN with bounded-precision neurons that can simulate any Turing Machine. We first describe how to encode the instantaneous description $(q, s_L, s_R) \in \mathcal{X}$ by a vector of rational numbers and two stacks, with which an RNN and its growing memory modules can be initialized. In the following discussion, we assume that all neurons have precision bounded by $p \geq 2$ in base $2|\Gamma|$; that is, each neuron can encode $p$ symbols at most.

Both the state $q$ and the top symbols $s_{L,(1)}, s_{R,(1)}$ are encoded with binary values as in Section 3. For the tape $s_j$ ($j \in \{L, R\}$), we define the fractal encoding $\rho^{(s)} : \Gamma^* \to \mathbb{Q}$ by:

$$\rho^{(s)}(y) := \sum_{i=1}^{|y|} \frac{y_{(i)}}{(2|\Gamma|)^i}, \tag{8}$$

which is similar to (3) except for the encoding of infinite blank symbols. Then, we encode the tape $s_j$ into a stack of neurons as follows: First, encode the rightmost $p$ symbols (the ones farthest from the read/write head) with $\rho^t$ and push it to an empty stack, denoted as $M_j$. Then, encode the next

rightmost $p$ symbols and push it to $M_j$ again, and repeat until at least one and at most $p$ symbols remain in the tape. Denote this encoding function for the tape as $\rho^{(M)} : \Gamma^* \to \mathbb{Q}^*$. The remaining symbols in the tape, denoted as $s_{j,(1:h(|s_j|))}$, where $h(y) := ((y - 1) \bmod p) + 1$, will be encoded with fractal encoding $\rho^{(s)}$ as well, but would appear in neurons inside the RNN.

The general idea is to let only the symbols closest to the read/write head $(s_{j,(1:h(|s_j|))})$ reside in the RNN. If the number of symbols residing in the RNN reaches $0$ or $p$, then we pop from or push to the stack respectively to ensure that at least $1$ and at most $p$ symbols reside in the tape neurons. It is interesting to note that the $k^{\text{th}}$ neuron in the stack (from the top) requires at least $kp$ steps of the Turing Machine before it may be updated, so the values of neurons near the bottom of the stack will not be changed for many steps. That is, the neurons in the stack, except for the top neurons, are passive.

*Example.* Let $\Gamma = \{1, 3, 5, 7\}$, $s_L = (3, 5, 7, 3, 5, 5, 3, 7)$ and $p = 3$. Then the number of symbols to remain in the RNN is $2$ and they are encoded by $\rho^{(s)}(s_{L,(1:h(|s_L|))}) = \frac{3}{8} + \frac{5}{8^2}$. The remaining six symbols are stored in the stack: $\rho^{(M)}(s_L) = [\frac{7}{8} + \frac{3}{8^2} + \frac{5}{8^3}, \frac{5}{8} + \frac{3}{8^2} + \frac{7}{8^3}]$.

We encode in neuron $h(|s_j|)$, the number of symbols in $s_j$ ($j \in \{L, R\}$). In the above example, $h(|s_L|) = 2$. These neurons help the RNN to know when pushing or popping operations are required. We use the encoding $\rho^{(h)} : \{1, 2, ..., p\} \to \mathbb{Q}$, defined by:

$$\rho^{(h)}(y) = \frac{y}{p + 1}. \tag{9}$$

Together, define the encoding function $\rho : \mathcal{X} \to (\mathbb{Q}^{2|\Gamma| + \lceil \log_2 |Q| \rceil + |Q||\Gamma| + 19}, \mathbb{Q}^*, \mathbb{Q}^*)$ by:

$$\rho(q, s_L, s_R) = (\rho^{(q)}(q) \oplus \rho^{(s)}(s_{L,(1:h(|s_L|))}) \oplus \rho^{(s)}(s_{R,(1:h(|s_R|))}) \oplus \rho^{(s)}(s_{L,(h(|s_L|)+1:h(|s_L|)+p)}) \oplus$$
$$\rho^{(s)}(s_{R,(h(|s_R|)+1:h(|s_R|)+p)}) \oplus \rho^{(r)}(s_{L,(1)}) \oplus \rho^{(r)}(s_{R,(1)}) \oplus \rho^{(h)}(h(|s_L|)) \oplus$$
$$\rho^{(h)}(h(|s_R|)) \oplus \mathbf{0}, \rho^{(M)}(s_L), \rho^{(M)}(s_R)), \tag{10}$$

where $\mathbf{0}$ is a zero vector of size $|Q||\Gamma| + 15$. The first element of the tuple is for initializing the neurons in the RNN, while the second and third element of the tuple is for initializing the two growing memory stack modules. All encoded values have precision of $p$. Similar to the previous section, $\rho$ is injective and so we can define the decoder function $\rho^{-1} : \rho(\mathcal{X}) \to \mathcal{X}$.

With the new encoding function $\rho$ and growing memory modules, we can prove an alternative version to Theorem 1 that only requires bounded precision neurons:

**Theorem 2.** *Given a Turing Machine $\mathcal{M}$, there exists an injective function $\rho : \mathcal{X} \to (\mathbb{Q}^n, \mathbb{Q}^*, \mathbb{Q}^*)$ and an $n$-neuron $p$-precision (in base $2|\Gamma|$) RNN with two growing memory modules $\mathcal{T}_{W,\mathbf{b}} : (\mathbb{Q}^n, \mathbb{Q}^*, \mathbb{Q}^*) \to (\mathbb{Q}^n, \mathbb{Q}^*, \mathbb{Q}^*)$, where $n = 2|\Gamma| + \lceil \log_2 |Q| \rceil + |Q||\Gamma| + 19$ and $p \geq 2$, such that for all instantaneous descriptions $x \in \mathcal{X}$,*

$$\rho^{-1}(\mathcal{T}_{W,\mathbf{b}}^3(\rho(x))) = \mathcal{P}_{\mathcal{M}}(x). \tag{11}$$

*Proof.* The detailed proof is in Appendix B. To illustrate the construction of the RNN, the parameters for the neuron initialized with $\rho^{(h)}(h(|s_L|))$, called the left guard neuron and denoted as $g_L(t)$, will be described here.

We assume all neurons in the RNN are initialized with values from the encoder function $\rho(x)$ at time $t = 1$. The guard neuron $g_L(t)$ encodes the number of left-tape symbols residing in the RNN. In three stages of an RNN, we need to update its value from $g_L(1) = h(|s_L|)/(p + 1)$ to $g_L(4) = h(|s_L'|)/(p + 1)$, where $s_L'$ is the left tape after one step of the Turing Machine. First, notice that $h(|s_L'|)$ can be expressed as:

$$h(|s_L'|) = \begin{cases} h(|s_L|) - 1 & \text{if } d = L \text{ and } h(|s_L|) \geq 2, \\ p & \text{if } d = L \text{ and } h(|s_L|) = 1, \\ h(|s_L|) + 1 & \text{if } d = R \text{ and } h(|s_L|) \leq p - 1, \\ 1 & \text{if } d = R \text{ and } h(|s_L|) = p, \end{cases} \tag{12}$$

where $d$ is the direction that the Turing Machine's head is moved.

*Example.* Assume $d = L$ and the size of the active symbols is $h(|s_L|) = 1$, which means that the Turing Machine is moving left and there is only one symbol residing in the RNN. Since after the move, $s_L$ will have one symbol less and so the corresponding neuron $s_L(t)$ will encode no symbols. As a result, the top neuron of the left growing memory module would be popped out, and $s_L(t)$ would assume its value. This way the neuron $s_L(t)$ again encodes the active symbols (1 to $p$) of the left tape series, and $h(|s_L|)$ is set to $p$. (Alternatively, one may prove it directly by the definition of $h$ and the fact that $|s'_L| = |s_L| - 1$.) An analogous process holds when the Turing Machine is moving right.

We implement (12) with an RNN as follows. Define stage neurons as:

$$c_1(t + 1) = \sigma(1 - c_1(t) - c_2(t)), \tag{13}$$
$$c_2(t + 1) = \sigma(c_1(t)), \tag{14}$$

with both neurons initialized to be zero. Define $\mathbf{c}(t) := [c_1(t), c_2(t), c_3(t)]$ where $c_3(t) := 1 - c_1(t) - c_2(t)$, then $\mathbf{c}(1) = [0, 0, 1]$; $\mathbf{c}(2) = [1, 0, 0]$; $\mathbf{c}(3) = [0, 1, 0]$. These stage neurons signal which one of the three stages that the RNN is in.

In the construction of the RNN, there exists a linear sum of neurons, denoted as $\mathbf{d}(t) = [d_L(t), d_R(t)]$, such that if the Turing Machine is moving left, $\mathbf{d}(1) = [0, 0]$; $\mathbf{d}(2) = [1, 0]$; $\mathbf{d}(3) = [0, 0]$; and if the Turing Machine is moving right, $\mathbf{d}(1) = [0, 0]$; $\mathbf{d}(2) = [0, 1]$; $\mathbf{d}(3) = [0, 0]$; this signals which direction the Turing Machine is moving to (formulas of $\mathbf{d}(t)$ appear in Appendix B).

Then consider the following update rule for the left guard neurons:

$$g_L(t + 1) = \sigma(g_L(t) + (d_R(t) - d_L(t) - pg'_L(t) + pg''_L(t))/(p + 1)), \tag{15}$$
$$g'_L(t + 1) = \sigma((p + 1)g_L(t) + d_R(t) - p - 2c_2(t) - 2c_3(t)), \tag{16}$$
$$g''_L(t + 1) = \sigma(2 - (p + 1)g_L(t) - d_R(t) - 2c_2(t) - 2c_3(t)), \tag{17}$$

where $g_L(1) = h(|s_L|)/(p+1)$, $g'_L(1) = g''_L(1) = 0$. It can be verified that $g_L(4) = h(|t'_L|)/(p+1)$ as defined by (12), completing the proof for $g_L(t)$.

*Example.* Assume $d = L$ and $h(|s_L|) = 1$. Then $g_L(1) = g_L(2) = 1/(p+1)$ and $g'_L(1) = g''_L(1) = g'_L(2) = g''_L(2) = 0$. On the second stage, $g_L(3) = \sigma(1/(p + 1) - 1/(p + 1)) = 0$, $g'_L(3) = \sigma(1 - p) = 0$, and $g''_L(3) = \sigma(2 - 1) = 1$. On the third stage, $g_L(4) = \sigma(0 + p/(p+1)) = p/(p+1)$ as required.

The full proof appears in Appendix B. $\square$

Similar to Corollary 1.1, it follows that:

**Corollary 2.1.** *Given a Turing Machine $\mathcal{M}$, there exists an injective function $\rho : \mathcal{X} \to (\mathbb{Q}^n, \mathbb{Q}^*, \mathbb{Q}^*)$ and an $n$-neuron $p$-precision (in base $|2\Gamma|$) RNN with two growing memory modules $\mathcal{T}_{W,\mathbf{b}} : (\mathbb{Q}^n, \mathbb{Q}^*, \mathbb{Q}^*) \to (\mathbb{Q}^n, \mathbb{Q}^*, \mathbb{Q}^*)$, where $n = 2|\Gamma| + \lceil \log_2 |Q| \rceil + |Q||\Gamma| + 19$ and $p \geq 2$, such that for all instantaneous descriptions $x \in \mathcal{X}$, the following holds: If $\mathcal{P}^*_{\mathcal{M}}(x)$ is defined, then*

$$\rho^{-1}(\mathcal{T}^*_{W,\mathbf{b}}(\rho(x))) = \mathcal{P}^*_{\mathcal{M}}(x), \tag{18}$$

*and if $\mathcal{P}^*_{\mathcal{M}}(x)$ is not defined, then $\mathcal{T}^*_{W,\mathbf{b}}(\rho(x))$ is also not defined. If $\mathcal{P}^*_{\mathcal{M}}(x)$ is defined and computed in $T$ steps by $\mathcal{M}$, then $\mathcal{T}^*_{W,\mathbf{b}}(\rho(x))$ is computed in $3T$ steps by the RNN.*

Finally, applying Corollary 2.1 to $U_{6,4}$, we obtain:

**Corollary 2.2.** *There exists a 54-neuron $p$-precision (in base 8) RNN with two growing memory modules that can simulate any Turing Machine in $\mathcal{O}(T^6)$, where $T$ is the number of steps required for the Turing Machine to compute the result and $p \geq 2$.*

The architecture of the Turing-complete 54-neuron RNN (fully described in the proof of Theorem 2) is depicted in Figure 2.

## 4.1 Relationship of the Growing Memory Modules with Stack-augmented RNNs

The proposed growing memory module belongs to the generic class of stack-augmented RNNs, which refers to any RNNs augmented with a stack-like mechanism. Many different forms of stack-augmented RNNs have been proposed [11, 12, 13, 14, 15]. Given the simplicity of the design,

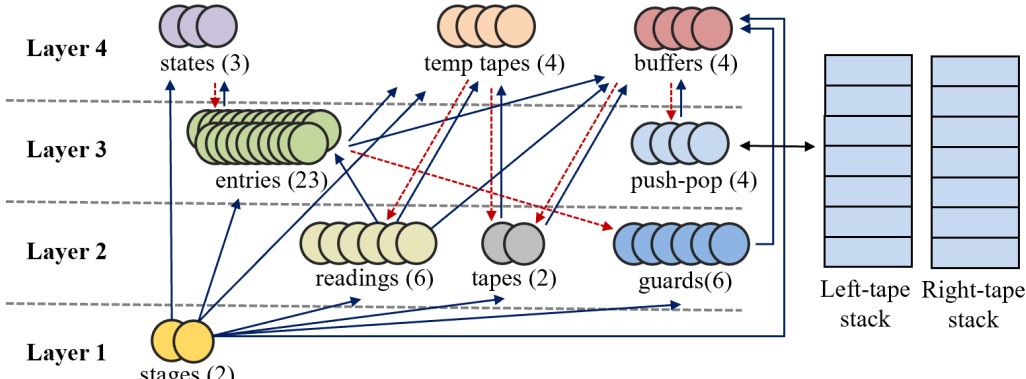

Figure 2: The architecture of the Turing-complete 54-neuron RNN with two growing memory modules. Blue (solid) lines denote bottom-up connections, and red (dotted) lines denote top-down connections. Neurons are grouped according to their respective functions. Notable neurons include: state neurons - represent the current state; tape neurons - represent the tape symbols near the read/write head using fractal encoding; guard neurons - represent the number of symbols residing in the RNN; push and pop neurons - control the pushing and popping of left-tape and right-tape stacks; stage neurons - represent the current stage and inhibits computation of other neurons if the neurons do not require updating in a stage. A detailed description of these neurons can be found in Appendix B.

the growing memory module represents the foundation form of these stack-augmented RNNs. It is easy to show that these stack-augmented RNNs can simulate the growing memory module in linear time. For example, the growing memory modules can be simulated by the neural stack [11] as follows. For pushing operation, set $v_t$ in the neural stack to $u(t)$ in the growing memory module, and $d_t$ in the neural stack to $1\{u(t) > 0\}$ in the growing memory module. For popping operation, set $u_t$ in the neural stack to $1\{o(t) = 1\}$ in the growing memory module. Therefore, the proof for the Turing completeness of bounded-precision RNNs with growing memory modules can extend to other stack-augmented RNNs. That is, bounded-precision stack-augmented RNNs are also Turing-complete.

Different from other stack-augmented RNNs, the proposed growing memory modules use a simple mechanism to control pushing and popping, as only the top neurons in the stack are included in the RNNs. This allows theories relating to growing memory modules to be easily extended to other forms of RNNs.

## 5 Bounded-Precision RNNs and Space-Bounded Turing Machines

As discussed above, it is inefficient to update all neurons that encode tape information, but if one still wants to remove the growing memory modules and focuses purely on a bounded-precision RNN only, then the resulting network can simulate space-bounded Turing Machines (that is, Turing Machines with a bounded-size tape) only.

**Theorem 3.** *Given a Turing Machine $\mathcal{M}$ with a bounded tape of size $F$, there exists an injective function $\rho : \mathcal{X} \to \mathbb{Q}^n$ and an $n$-neuron $p$-precision (in base $|2\Gamma|$) RNN $\mathcal{T}_{W,\mathbf{b}} : \mathbb{Q}^n \to \mathbb{Q}^n$, where $n = \mathcal{O}(\lceil F/p \rceil)$ and $p \geq 2$, such that for all instantaneous descriptions $x \in \mathcal{X}$,*

$$\rho^{-1}(\mathcal{T}^3_{W,\mathbf{b}}(\rho(x))) = \mathcal{P}_{\mathcal{M}}(x). \tag{19}$$

The proof, which is constructive, can be found in Appendix C. The general idea of the proof is to implement the growing memory module in Section 4 by an RNN as well and place all neurons inside the RNN. The theorem shows that the number of neurons required to simulate a space-bounded Turing Machine correlates with the tape size.

To simulate a Turing Machine with an unbounded tape, we would need to add neurons to the RNN once the read/write head reaches the end of the memory. To be specific, we say an RNN has an

unbounded number of neurons if the RNN either has an infinite number of neurons to start with or can increase the number of neurons after each update step depending on the neurons' values. The UTM that simulates any Turing Machine in a fast way was described in [21], which does so with only $\mathcal{O}(T \log T)$ slowdown. While this UTM has multiple tapes, Theorem 3 can be generalized to multiple-tape Turing Machines easily. We now obtain:

**Corollary 3.1.** *There exists an unbounded-neuron bounded-precision RNN that can simulate any Turing Machine in $\mathcal{O}(T \log T)$, where $T$ is the number of steps required for the Turing Machine to compute the result.*

## 6  Discussion and Conclusion

To construct a Turing-complete RNN, we have to incorporate some encoding for the unbounded number of symbols on the Turing tape. This encoding can be done by: (a) unbounded precision of some neurons (Theorem 1), (b) an unbounded number of neurons (Theorem 3), or (c) a separate growing memory module (Theorem 2). The main contribution of this paper is spelling out the details of (c), which provides a practical way to construct an RNN that runs any given algorithms. We prove the Turing completeness of a 40-neuron unbounded-precision RNN, which is the smallest Turing-complete RNN to date. We analyze the relationship between the number of neurons and the precision of an RNN when simulating a Turing Machine. Most importantly, we propose a 54-neuron bounded-precision RNN with growing memory modules that is Turing-complete, and this proof of Turing completeness can be extended to stack-augmented RNNs in general.

This paper focuses on the computational capabilities and representability of symbolic and sub-symbolic processing via stack-augmented RNNs; it does not engage yet with methods to train them. Since growing memory modules are not differentiable, we cannot train them directly by the frequently used error backpropagation algorithm. One may want to construct a differentiable version of the modules, or alternatively use a different learning rule (e.g., REINFORCE [22]) to deal with the discrete pushing and popping operations. Understanding methods to train growing memory modules, incorporate symbolic information into the sub-symbolic representation efficiently, and retrieve both symbolic and non-symbolic information are the next steps towards the goal of combining symbolic and sub-symbolic capabilities in an adaptive and applicable manner.

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
