# Supplementary for Turing Completeness of Bounded-Precision Recurrent Neural Networks

**Stephen Chung**[*]
Department of Computer Science
University of Massachusetts Amherst
Amherst, MA 01003
minghaychung@umass.edu

**Hava Siegelmann**[*]
Department of Computer Science
University of Massachusetts Amherst
Amherst, MA 01003
hava@umass.edu

## A Proof of Theorem 1

*Proof.* Without loss of generality, assume $Q = \{1, 2, ..., |Q|\}$ and $\Gamma = \{1, 3, 5, ..., 2|\Gamma| - 1\}$. Also, let $x = (q, s_L, s_R)$ and $\mathcal{P}_\mathcal{M}(x) = (q', s'_L, s'_R)$ (not to be confused with neurons' values which are function of $t$). Let $\rho : \mathcal{X} \to \mathbb{Q}^n$ be the function described by (5) and $\mathbf{v}(t) \in \mathbb{Q}^n$ denotes the neurons' value of an RNN $\mathcal{T}_{W,\mathbf{b}}$ at time $t$. Initialize $\mathbf{v}(1) = \rho(x)$. It suffices to prove that there exists $\mathcal{T}_{W,\mathbf{b}}$ such that $\mathbf{v}(4) = \rho(\mathcal{P}_\mathcal{M}(x))$.

The RNN $\mathcal{T}_{W,\mathbf{b}}$ is constructed as follows. The $n = 2|\Gamma| + \lceil \log_2 |Q| \rceil + |Q||\Gamma| + 5$ neurons are classified into six groups:

1. Stage neurons $\mathbf{c}(t) \in \{0,1\}^2$, initialized with $\mathbf{c}(1) = \mathbf{0}$;

2. Entry neurons $\mathbf{e}(t) \in \{0,1\}^{|Q||\Gamma|-1}$, initialized with $\mathbf{e}(1) = \mathbf{0}$;

3. Temporary tape neurons $s'_j(t), s''_j(t) \in \mathbb{Q}$, initialized with $s'_j(1) = s''_j(1) = 0$, where $j \in \{L, R\}$;

4. Tape neurons $s_j(t) \in \mathbb{Q}$, initialized with $s_j(1) = \rho^{(s)}(s_j)$, where $j \in \{L, R\}$;

5. Readout neurons $\mathbf{r}_j(t) \in \{0,1\}^{|\Gamma|-1}$, initialized with $\mathbf{r}_j(1) = \rho^{(r)}(s_{j,(1)})$, where $j \in \{L, R\}$;

6. State neurons $\mathbf{q}(t) \in \{0,1\}^{\lceil \log_2 |Q| \rceil}$, initialized with $\mathbf{q}(1) = \rho^{(q)}(q)$.

Before describing the update rules for these neurons, we define three linear functions $readQ, readSO$ and $readSF$ that facilitates reading the encoding of neurons:

Define $readQ : \{0,1\}^{\lceil \log_2 |Q| \rceil} \to \{0,1\}^{|Q|}$ by:

$$readQ_i(\mathbf{y}) = (2\rho^{(q)}(i) - 1) \cdot (2\mathbf{y} - 1) - (\lceil \log_2 |Q| \rceil - 1), \tag{20}$$

where $i \in Q$ and $\cdot$ denotes the dot product. This function transforms the value of state neurons into one-hot encoding of state, since $readQ_n(\rho^{(q)}(n)) = 1$ and $readQ_i(\rho^{(q)}(n)) \le 0$ for $i \ne n$.

*Example.* Using the same example when defining $\rho^{(q)}$ in Section 3, we have $readQ_i(\rho^{(q)}(0)) = [+1, -1, -1, , -3, -1, -3]$, $readQ_i(\rho^{(q)}(1)) = [-1, +1, -3, -1, -3, -1]$, $readQ_i(\rho^{(q)}(2)) = [-1, -3, +1, -1, -3, -5]$, etc.

Define $readSO : \{0,1\}^{|\Gamma|-1} \to \{0,1\}^{|\Gamma|}$ by:

$$readSO_j(\mathbf{y}) = ([1] \oplus \mathbf{y} \oplus [0])_j - ([1] \oplus \mathbf{y} \oplus [0])_{j+1}, \tag{21}$$

[*]Both authors contributed equally.

35th Conference on Neural Information Processing Systems (NeurIPS 2021).

where $j \in \{1, 2, ..., |\Gamma|\}$. This function transforms the value of readout neurons into the one-hot encoding of symbols.

*Example.* Assume $|\Gamma| = 4$ and $\mathbf{r}_L = [1, 1, 0]$, then it means that the first symbol of $s_L$ is 5 (since $\rho^{(r)}(5) = [1, 1, 0]$) and $readSO(\mathbf{r}_L) = [0, 0, 1, 0]$.

Define $readSF : \{0, 1\}^{|\Gamma|-1} \to \mathbb{Q}$ by:

$$readSF(\mathbf{y}) = \frac{1 + 2\sum_{k=1}^{|\Gamma|-1} \mathbf{y}_k}{2|\Gamma|}. \tag{22}$$

This function transforms the value of readout neurons into the fractal encoding of symbols.

*Example.* Assume $|\Gamma| = 4$ and $\mathbf{r}_L = [1, 1, 0]$, then it means the first symbol of $s_L$ is 5 (since $\rho^{(r)}(5) = [1, 1, 0]$) and $readSF(\mathbf{r}_L) = \frac{5}{8}$.

The update rule for each group of neurons are described next:

*1. Stage neurons.*

$$\mathbf{c}_1(t + 1) = \sigma(1 - \mathbf{c}_1(t) - \mathbf{c}_2(t)), \tag{23}$$
$$\mathbf{c}_2(t + 1) = \sigma(\mathbf{c}_1(t)). \tag{24}$$

We also denote $\mathbf{c}_3'(t) = 1 - \mathbf{c}_1(t) - \mathbf{c}_2(t)$, which is not a neuron but a linear sum of neuron. Let $\mathbf{c}'(t) = [\mathbf{c}_1(t), \mathbf{c}_2(t), \mathbf{c}_3'(t)]$. It follows that $\mathbf{c}'(1) = [0, 0, 1]$; $\mathbf{c}'(2) = [1, 0, 0]$; $\mathbf{c}'(3) = [0, 1, 0]$; $\mathbf{c}'(4) = [0, 0, 1]$. $\mathbf{c}'(t)$ thus signals which one of the three steps that the RNN is in.

*2. Entry neurons.*

$$\mathbf{e}_{i,j}(t + 1) = \sigma(readQ_i(\mathbf{q}(t)) + readSO_j(\mathbf{r}_L(t)) - 1 - \mathbf{c}_1'(t) - \mathbf{c}_2'(t)), \tag{25}$$

where $i, j \in Q \times \Gamma - \{(|Q|, 2|\Gamma| - 1)\}$. Let $\mathbf{e}_{|Q|,2|\Gamma|-1}'(t) = \mathbf{c}_1'(t) - \sum_{i,j \in Q \times \Gamma - \{|Q|,2|\Gamma|-1\}} \mathbf{e}_{i,j}(t)$, which is not a neuron but a linear sum of neuron that represents the last combination ($|Q|, 2|\Gamma| - 1$). Also let $\mathbf{e}'(t) = \mathbf{e}(t) \oplus [\mathbf{e}_{|Q|,2|\Gamma|-1}'(t)]$.

It follows that $\mathbf{e}_{i,j}'(2) = 1$ if the state of the Turing Machine is $i$ and the top left-tape symbol is $j$, and 0 otherwise. Note that $\mathbf{e}'(1) = \mathbf{e}'(3) = \mathbf{e}'(4) = \mathbf{0}$. $\mathbf{e}'(t)$ thus gives a one-hot encoding of the combination of the state and the top left-tape symbol at $t = 2$.

Since the combination of the state and the top left-tape symbol fully determines the next transition of the Turing Machine, we can use the entry neurons to determine the next transition. Define a linear sum of entry neurons as follows:

$$move_L(t) = \sum_{k,h \in \mathcal{L}} \mathbf{e}_{k,h}'(t), \tag{26}$$

$$move_R(t) = \sum_{k,h \in \mathcal{R}} \mathbf{e}_{k,h}'(t), \tag{27}$$

$$enter_i(t) = \sum_{k,h \in \mathcal{W}_i} \mathbf{e}_{k,h}'(t), \tag{28}$$

$$write_j(t) = \sum_{k,h \in \mathcal{E}_j} \mathbf{e}_{k,h}'(t), \tag{29}$$

where $i \in Q, j \in \Gamma$, $\mathcal{L} = \{k, h \in Q \times \Gamma : \delta_3(k, h) = L\}$ (i.e. all possible combination of states and symbols that leads to moving left), $\mathcal{R} = \{k, h \in Q \times \Gamma : \delta_3(k, h) = R\}$ (i.e. all possible combination of states and symbols that leads to moving right), $\mathcal{W}_i = \{k, h \in Q \times \Gamma : \delta_1(k, h) = i\}$ (i.e. all possible combination of states and symbols that leads to entering state $i$) and $\mathcal{E}_j = \{k, h \in Q \times \Gamma : \delta_2(k, h) = j\}$ (i.e. all possible combination of states and symbols that leads to writing symbol $j$).

It follows that: i. $move_L(2) = 1$ if the Turing Machine is moving left, and 0 otherwise; ii. $move_R(2) = 1$ if the Turing Machine is moving right, and 0 otherwise; iii. $enter_i(2) = 1$ if the Turing Machine is entering state $i$, and 0 otherwise; iv. $write_j(2) = 1$ if the Turing Machine is writing symbol $j$, and 0 otherwise; v. For $t \in \{1, 3, 4\}$, $move_L(t) = move_R(t) = enter_i(t) = write_j(t) = 0$ for all $i \in Q, j \in \Gamma$.

*3. Temporary tape neurons.*

$$s'_L(t+1) = \sigma(2|\Gamma|(s_L(t) - readSF(\mathbf{r}_L(t)) - move_R(t) - \mathbf{c}'_2(t) - \mathbf{c}'_3(t))), \tag{30}$$

$$s''_L(t+1) = \sigma(readSF(\mathbf{r}_R(t)) + (2|\Gamma|)^{-1}(s_L(t) - readSF(\mathbf{r}_L(t)) + writeSF(t)) -$$
$$move_L(t) - \mathbf{c}'_2(t) - \mathbf{c}'_3(t)), \tag{31}$$

$$s'_R(t+1) = \sigma((2|\Gamma|)^{-1}s_R(t) + writeSF(t) - move_R(t) - \mathbf{c}'_2(t) - \mathbf{c}'_3(t)), \tag{32}$$

$$s''_R(t+1) = \sigma(2\Gamma(s_R(t) - readSF(\mathbf{r}_R(t)) - move_L(t) - \mathbf{c}'_2(t) - \mathbf{c}'_3(t))), \tag{33}$$

where $writeSF(t) = \sum_{j\in\Gamma} j(2|\Gamma|)^{-1}write_j(t)$, which is the fractal encoding for the symbol to be written. The temporary tape neurons compute the new value of the left tape and the right tape depending on the moving direction of the Turing Machine. The common term $-\mathbf{c}'_2(t) - \mathbf{c}'_3(t)$ ensures that temporary tape neurons equal 0 at $t \in \{2, 4\}$. Detailed explanation of these neurons are as follows:

$s'_L(3) = \rho^{(s)}(s'_L)$ if the Turing Machine is moving left, and 0 otherwise: If the Turing Machine is moving left, the top left-tape symbol has to be removed from the left tape, which can be done by subtracting the top left-tape symbol $readSF(\mathbf{r}_L(t))$. Then, the left tape has to be shifted right, which can be done by multiplying the left tape by $2|\Gamma|$. The term $-move_R(t)$ ensures that $s'_L(3) = 0$ if the Turing Machine is moving right.

$s''_L(3) = \rho^{(s)}(s'_L)$ if the Turing Machine is moving right, and 0 otherwise: If the Turing Machine is moving right, the top left-tape symbol has to be replaced by the symbol to be written, which can be done by subtracting $readSF(\mathbf{r}_L(t))$ and adding $writeSF(t)$. Then the left tape has to be shifted left, which can be done by multiplying the left tape by $(2|\Gamma|)^{-1}$ and adding the top right-tape symbol, $readSF(\mathbf{r}_R(t))$, to it. The term $-move_L(t)$ ensures that $s''_L(3) = 0$ if the Turing Machine is moving left.

$s'_R(3) = \rho^{(s)}(s'_R)$ if the Turing Machine is moving left, and 0 otherwise: If the Turing Machine is moving left, the right tape has to be shifted right, which can be done by multiplying the right tape by $(2|\Gamma|)^{-1}$. Then, a new symbol also has to be written to the top of the right tape, which can be done by adding $writeSF(t)$ to the right tape. The term $-move_R(t)$ ensures that $s'_R(3) = 0$ if the Turing Machine is moving right.

$s''_R(3) = \rho^{(s)}(s'_R)$ if the Turing Machine is moving right, and 0 otherwise: If the Turing Machine is moving right, the top right-tape symbol has to be removed from the right tape, which can be done by subtracting the top right-tape symbol $readSF(\mathbf{r}_R(t))$. Then, the right tape has to be shifted left, which can be done by multiplying the right tape by $2|\Gamma|$. The term $-move_L(t)$ ensures that $s''_R(3) = 0$ if the Turing Machine is moving left.

Together, we have $s'_L(3) + s''_L(3) = \rho^{(s)}(s'_L)$ and $s'_R(3) + s''_R(3) = \rho^{(s)}(s'_R)$, ready to be used to update $s_L(4)$ and $s_R(4)$.

*4. Tape neurons.*

$$s_L(t+1) = \sigma(s_L(t) + s'_L(t) + s''_L(t) - \mathbf{c}'_1(t)), \tag{34}$$

$$s_R(t+1) = \sigma(s_R(t) + s'_R(t) + s''_R(t) - \mathbf{c}'_1(t)). \tag{35}$$

It follows that $s_L(1) = s_L(2) = \rho^{(s)}(s_L)$, $s_L(3) = 0$ and $s_L(4) = \rho^{(s)}(s'_L)$; $s_R(1) = s_R(2) = \rho^{(s)}(s_R)$, $s_R(3) = 0$ and $s_R(4) = \rho^{(s)}(s'_R)$.

*5. Readout neurons.*

$$\mathbf{r}_{j,i}(t+1) = \sigma((2i+1)\mathbf{r}_{j,i}(t) + 2|\Gamma|(s'_j(t) + s''_j(t)) - 2i - \mathbf{c}'_1(t)), \tag{36}$$

where $j \in \{L, R\}$ and $i \in \{1, 2, ..., |\Gamma| - 1\}$. Note that $2|\Gamma|(s'_j(3) + s''_j(3)) - 2i \geq 1$ if the top symbol in $s'_j$ is larger than $2i$, and $\leq 0$ otherwise. It follows that $\mathbf{r}_j(1) = \mathbf{r}_j(2) = \rho^{(r)}(s_{j,(1)})$, $\mathbf{r}_j(3) = \mathbf{0}$ and $\mathbf{r}_j(4) = \rho^{(r)}(s'_{j,(1)})$.

*6. State neurons.*

$$\mathbf{q}_i(t+1) = \sigma\left(\mathbf{q}_i(t) + \left(\sum_{k\in Q} \rho_i^{(q)}(k)enter_k(t)\right) - \mathbf{c}'_3(t)\right), \tag{37}$$

where $i \in \{1, 2, ..., \lceil \log_2 |Q| \rceil\}$. Note that $enter_k(2) = 1$ if $q' = k$ and 0 otherwise. It follows that $\mathbf{q}(1) = \rho^{(q)}(q)$, $\mathbf{q}(2) = \mathbf{0}$, $\mathbf{q}(3) = \mathbf{q}(4) = \rho^{(q)}(q')$.

Together, $\mathbf{v}(4) = \rho(\mathcal{P}_\mathcal{M}(x))$. This completes the proof. $\qquad\square$

## B   Proof of Theorem 2

*Proof.* The proof is similar to that of Theorem 1 but with more neurons. Without loss of generality, assume $Q = \{1, 2, ..., |Q|\}$ and $\Gamma = \{1, 3, 5, ..., 2|\Gamma| - 1\}$. Also, let $x = (q, s_L, s_R)$ and $\mathcal{P}_\mathcal{M}(x) = (q', s'_L, s'_R)$ (not to be confused with neurons' values which are function of $t$). Let $\rho : \mathcal{X} \rightarrow (\mathbb{Q}^n, \mathbb{Q}^*, \mathbb{Q}^*)$ be the function described by (10). Consider an RNN with two growing memory modules $\mathcal{T}_{W,\mathbf{b}}$. Let $\mathbf{v}(t) \in \mathbb{Q}^n$ denotes the neurons' value of the RNN at time $t$, and let $M_L(t) \in \mathbb{Q}^*$, $M_R(t) \in \mathbb{Q}^*$ denote the two stacks' values at time $t$. Initialize $(\mathbf{v}(1), M_L(1), M_R(1)) = \rho(x)$. It suffices to prove that there exists $\mathcal{T}_{W,\mathbf{b}}$ such that $(\mathbf{v}(4), M_L(4), M_R(4)) = \rho(\mathcal{P}_\mathcal{M}(x))$.

Define $\rho_0^{(s)} : \mathbb{Q}^* \rightarrow \mathbb{Q}$ by $\rho_0^{(s)}(y) = \rho^{(s)}(y_{(1:h(|y|))})$ and $\rho_1^{(s)} : \mathbb{Q}^* \rightarrow \mathbb{Q}$ by: $\rho_1^{(s)}(y) = \rho^{(s)}(y_{(h(|y|)+1:h(|y|)+p)})$. $\rho_0^{(s)}(s_j)$ represents the symbols residing in the tape neuron of the RNN while $\rho_1^{(s)}(s_j)$ represents the following $p$ symbols, which is also the top neuron's value in the stack $M_j(1)$.

The RNN $\mathcal{T}_{W,\mathbf{b}}$ is constructed as follows. The $n = 2|\Gamma| + \lceil \log_2 |Q| \rceil + |Q||\Gamma| + 19$ neurons are classified into nine groups:

1. Stage neurons $\mathbf{c}(t) \in \{0, 1\}^2$, initialized with $\mathbf{c}(1) = \mathbf{0}$;

2. Entry neurons $\mathbf{e}(t) \in \{0, 1\}^{|Q||\Gamma|-1}$, initialized with $\mathbf{e}(1) = \mathbf{0}$;

3. Temporary tape neurons $s'_j(t), s''_j(t) \in \mathbb{Q}$, initialized with $s'_j(1) = s''_j(1) = 0$, where $j \in \{L, R\}$;

4. Tape neurons $s_j(t) \in \mathbb{Q}$, initialized with $s_j(1) = \rho_0^{(s)}(s_j)$, where $j \in \{L, R\}$;

5. Readout neurons $\mathbf{r}_j(t) \in \{0, 1\}^{|\Gamma|-1}$, initialized with $\mathbf{r}_j(1) = \rho^{(r)}(s_{j,(1)})$, where $j \in \{L, R\}$;

6. State neurons $\mathbf{q}(t) \in \{0, 1\}^{\lceil \log_2 |Q| \rceil}$, initialized with $\mathbf{q}(1) = \rho^{(q)}(q)$;

7. Guard neurons $g_j(t), g'_j(t), g''_j(t) \in \mathbb{Q}$, initialized with $g_j(1) = \rho^{(h)}(h(|s_j|))$ and $g'_j(1) = g''_j(1) = 0$, where $j \in \{L, R\}$;

8. Buffer neurons $\beta_j(t), \beta'_j(t) \in \mathbb{Q}$, initialized with $\beta_j(1) = \beta'_j(1) = 0$, where $j \in \{L, R\}$;

9. Push-pop neurons $o_j(t), u_j(t) \in \mathbb{Q}$, initialized with $o_j(1) = \rho_1^{(s)}(s_j)$, $u_j(1) = 0$, where $j \in \{L, R\}$.

The general idea of the proof is that the required update can be constructed as a two-step process. In the first step, we apply the equations used in the proof of Theorem 1 for neurons from 1. to 6. There are three cases for the second step:

i. If the updated left-tape neuron holds $1 \leq y \leq p$ symbols: no pushing or popping is required, and both the left-tape neuron and the left-tape stack do not require further update;

ii. if the updated left-tape neuron holds $0$ symbols: it is required to pop the top $p$ symbols from the left-tape stack, and the popped symbols have to be added to the left-tape neuron. The left-tape readout neuron also has to be updated to the encoding for the top symbol in the new left-tape neuron's value;

iii. if the updated left-tape neuron holds $p + 1$ symbols, then it is required to push the bottom $p$ symbols of it to the left-tape stack, and the pushed symbols have to be removed from the left-tape neuron.

A similar process holds for the right tape. Therefore, the equations for neurons from 1. to 6. are almost the same as the one used in the proof of Theorem 1. In the following proof, we use the same notation as in the proof of Theorem 1.

First, to determine whether pushing or popping is required, the number of symbols in the left-tape ($h(|s_L|)$) and right-tape neurons ($h(|s_R|)$) have to be kept track of. This is done by the guard neurons:

*7. Guard neurons.*

$$g_j(t+1) = \sigma(g_L(t) + (move_{\neg j}(t) - move_j(t) - pg_j'(t) + pg_j''(t))/(p+1)), \qquad (38)$$

$$g_j'(t+1) = \sigma((p+1)g_j(t) + move_{\neg j}(t) - p - 2\mathbf{c}_2(t) - 2\mathbf{c}_3'(t)), \qquad (39)$$

$$g_j''(t+1) = \sigma(2 - (p+1)g_j(t) - move_{\neg j}(t) - 2\mathbf{c}_2(t) - 2\mathbf{c}_3'(t)), \qquad (40)$$

where $j \in \{L, R\}$ and $\neg j$ denotes the opposite direction of $j$. The explanation is in the main paper and is not repeated here (the equations are slightly different from the one shown in the main paper because we use the notation defined in the proof of Theorem 1 here). It follows that $g_j(1) = h(|s_j|)/(p+1)$, $g_j'(1) = g_j''(1) = 0$, $g_j(4) = h(|s_j|)/(p+1)$ and $g_j'(4) = g_j''(4) = 0$.

*8. Buffer neurons.*

$$\beta_L(t+1) = \sigma(o_L(t) - move_R(t) - (p+1)g_L(t) + 1 - 2\mathbf{c}_2'(t) - 2\mathbf{c}_3'(t)), \qquad (41)$$

$$\beta_L'(t+1) = \sigma(s_L(t) - readSF(\mathbf{r}_L(t)) + writeSF(t) - move_L(t) - p + (p+1)g_L(t) - \\ 2\mathbf{c}_2'(t) - 2\mathbf{c}_3'(t)), \qquad (42)$$

$$\beta_R(t+1) = \sigma(o_R(t) - move_L(t) - (p+1)g_R(t) + 1 - 2\mathbf{c}_2'(t) - 2\mathbf{c}_3'(t)), \qquad (43)$$

$$\beta_R'(t+1) = \sigma(s_R(t) - move_R(t) - p + (p+1)g_R(t) - 2\mathbf{c}_2'(t) - 2\mathbf{c}_3'(t)), \qquad (44)$$

where $readSF$ and $writeSF$ are defined the same as in the proof of Theorem 1. The buffer neurons $\beta_j(t)$ compute the new value to be popped while $\beta_j'(t)$ compute the values to be pushed. The common terms $-2\mathbf{c}_2'(t) - 2\mathbf{c}_3'(t)$ ensure that the buffer neurons equal 0 at $t \in \{2, 4\}$. Detailed explanation of these neurons are as follows:

$\beta_L(3) = \rho_1^{(s)}(s_L)$ if popping is required for the left stack, and 0 otherwise: If $move_R(2) = 0$ and $g_L(2) = 1/(p+1)$, it means that the left-tape neuron is currently holding 1 symbols and the Turing Machine is moving left, implying that the left-tape neuron will hold 0 symbols and popping is required. In this case, $-move_R(2) - (p+1)g_L(2) + 1 = 0$, so $\beta_L(3) = o_L(2) = \rho_1^{(s)}(s_L)$, which is the value to be popped (the neuron's value on the top of the left-tape stack). In other cases, $-move_R(2) - (p+1)g_L(2) + 1 \le -1$ so $\beta_L(3) = 0$.

$\beta_L'(3) = \rho_1^{(s)}(s_L')$ if pushing is required for the left stack, and 0 otherwise: If $move_L(2) = 0$ and $g_L(2) = p/(p+1)$, it means that the left-tape neuron is currently holding $p$ symbols and the Turing Machine is moving right, implying that the left-tape neuron will hold $p + 1$ symbols and pushing is required. In this case, $-move_L(2) - p + (p+1)g_L(2) = 0$, so $\beta_L'(3) = s_L(2) - readSF(\mathbf{r}_L(2)) + writeSF(2)$, which is top $p$ symbols of the left tape but with the top symbol updated. This is also the value to be pushed, $\rho_1^{(s)}(s_L')$. In other cases, $-move_L(2) - p + (p+1)g_L(2) \le -1$ so $\beta_L'(3) = 0$.

$\beta_R(3) = \rho_1^{(s)}(s_R)$ if popping is required for the right stack, and 0 otherwise: the explanation is similar to $\beta_L(t)$ and is omitted here.

$\beta_R'(3) = \rho_1^{(s)}(s_R')$ if pushing is required for the right stack, and 0 otherwise: the explanation is similar to $\beta_L'(t)$. However, the value to be pushed is just $s_R(t)$ instead of $s_L(t) - readSF(\mathbf{r}_L(t)) + writeSF(t)$, since the top right-tape symbol does not require updating.

*9. Push-pop neurons.*

$$o_j(t+1) = \sigma(o_j(t) - \beta_j(t)), \qquad (45)$$

$$u_j(t+1) = \sigma(u_j(t) + \beta_j'(t)), \qquad (46)$$

where $j \in \{L, R\}$. $o_j(t)$ is the pop neuron for the stack $M_j(t)$ and $u_j(t)$ is the push neuron for the stack $M_j(t)$.

For the pop neuron, it can be seen that $o_j(1) = o_j(2) = o_j(3) = \rho_1^{(s)}(s_j)$. At $t = 4$, if popping is required, $\beta_j(3) = \rho_1^{(s)}(s_j)$ hence $o_j(4)$ will be set to 0 by the RNN. Then, due to the mechanism

of the growing memory module, the top neuron in $M_j(t)$ will be popped from the stack and $o_j(4)$ will be set to the next top neuron, which is $\rho_1^{(s)}(s_j')$. Otherwise, if popping is not required, then $o_j(4) = \rho_1^{(s)}(s_j) = \rho_1^{(s)}(s_j')$ and the stack remain unchanged. Note that the default popping value $c$ in Definition 3 is set to the fractal encoding for $p$ blank symbols, so the update is still valid if no neurons remain in the stack.

For the pushing neuron, it can be seen that $u_j(1) = u_j(2) = u_j(3) = 0$. At $t = 4$, if pushing is required, $\beta_j'(3) = \rho_1^{(s)}(s_j')$ hence $u_j(4)$ will be set to $\rho_1^{(s)}(s_j')$ by the RNN. Then, due to the mechanism of the growing memory module, a new neuron with value $\rho_1^{(s)}(s_j')$ will be pushed to the stack and $u_j(4)$ will be set to 0. Otherwise, if pushing is not required, then $u_j(4) = 0$ and the stack remain unchanged.

Through the above operations, the stacks are updated to the desired value at time $t = 4$. That is, $M_j(1) = M_j(2) = M_j(3) = \rho^{(M)}(s_j)$ and $M_j(4) = \rho^{(M)}(s_j')$ for $j \in \{L, R\}$.

The update equations for 1. to 6. are the same as that in the proof of Theorem 1, except:

*4. Tape neurons.*

$$s_L(t + 1) = \sigma(s_L(t) + s_L'(t) + s_L''(t) + \beta_L(t) - (2|\Gamma|)^{-1}\beta_L'(t) - \mathbf{c}_1'(t)), \qquad (47)$$
$$s_R(t + 1) = \sigma(s_R(t) + s_R'(t) + s_R''(t) + \beta_R(t) - (2|\Gamma|)^{-1}\beta_R'(t) - \mathbf{c}_1'(t)). \qquad (48)$$

Different from the update equations in Theorem 1, we add the term $\beta_j(t) - (2|\Gamma|)^{-1}\beta_j'(t)$. Adding $\beta_j(t)$ ensures that when popping is required, the value popped from the stack (that is, the top $p$ symbols in the stack) is added to the tape neuron. Similarly, subtracting $(2|\Gamma|)^{-1}\beta_j'(t)$ ensures that when pushing is required, the bottom $p$ symbols in the tape neurons are removed. It follows that $s_j(1) = s_j(2) = \rho_0^{(s)}(s_j)$, $s_j(3) = 0$, and $s_j(4) = \rho_0^{(s)}(s_j')$.

*6. Readout neurons.*

$$\mathbf{r}_{j,i}(t + 1) = \sigma((2i + 1)\mathbf{r}_{j,i}(t) + 2|\Gamma|(s_j'(t) + s_j''(t) + \beta_j(t)) - 2i - \mathbf{c}_1'(t)), \qquad (49)$$

where $j \in \{L, R\}$ and $i \in \{1, 2, ..., |\Gamma| - 1\}$. Different from the update equations in Theorem 1, we add the term $\beta_j(t)$. This is because if popping is required, $s_j'(t) + s_j''(t)$ equals 0 and the top symbol of the updated tape resides in $\beta_j(t)$ instead, so we need to read from $\beta_j(t)$ instead of $s_j'(t) + s_j''(t)$. It follows that $\mathbf{r}_j(1) = \mathbf{r}_j(2) = \rho^{(r)}(s_{j,(1)})$, $\mathbf{r}_j(3) = \mathbf{0}$ and $\mathbf{r}_j(4) = \rho^{(r)}(s_{j,(1)}')$.

Together, $(\mathbf{v}(4), M_L(4), M_R(4)) = \rho(\mathcal{P}_{\mathcal{M}}(x))$. This completes the proof. $\qquad\square$

## C  Proof of Theorem 3

*Proof.* The proof is similar to that of Theorem 2 but with more neurons. Without loss of generality, assume $Q = \{1, 2, ..., |Q|\}$ and $\Gamma = \{1, 3, 5, ..., 2|\Gamma| - 1\}$. Also, let $x = (q, s_L, s_R)$ and $\mathcal{P}_{\mathcal{M}}(x) = (q', s_L', s_R')$ (not to be confused with neurons' values which are function of $t$). Assume that the blank symbols are not truncated in $s_L$ or $s_R$ and that the tape's size is $F$, i.e. $|s_L| + |s_R| = F$. Also, let $f = \lceil F/p \rceil$.

The encoding function $\rho$ is constructed as follows. Define $\rho^{(M)} : Q^* \to Q^f$ by:

$$\rho_i^{(M)}(y) := \begin{cases} \rho^{(s)}(y_{(-(i-1)p-1:-ip)}), & \text{if } i < |y|/p, \\ 0, & \text{else,} \end{cases} \qquad (50)$$

where $i \in \{1, 2, ..., f\}$ and $s_{(-j:-k)}$ denotes $s_{(|s|-k+1)}s_{(|s|-k+2)}...s_{(|s|-j+1)}$ for $k > j > 0$. This function encodes the tape into a fixed-size vector with each neuron holding $p$ symbols.

*Example.* Let $p = 4$, $|\Gamma| = 4$ and $s_L = (5, 3, 1, 3, 7, 3, 3, 7, 1, 3, 5)$ (recall that the leftmost symbol is the closest symbol to the read/write head in the representation of $s_L$). Then it follows $\rho^{(M)}(s_L) = [\rho^{(s)}(7, 1, 3, 5), \rho^{(s)}(3, 7, 3, 3), 0, 0, ..., 0]$. It is almost the same as the $\rho^{(M)}(t)$ defined in the proof of Theorem 2 but with zero appended after it, so the size of the output vector is always $f$. $\rho^{(M)}$ is used to encode both the left tape and the right tape.

Also, define $\rho^{(d)} : Q^* \to \{0, 1\}^f$ by:

$$\rho_i^{(d)}(y) := \begin{cases} 1, & \text{if } i = \lceil |y|/p - 1 \rceil, \\ 0, & \text{else,} \end{cases} \tag{51}$$

where $i \in \{1, 2, ..., f\}$. This function encodes the position of the last non-zero element in $\rho^{(M)}(t)$. That is, the position with the last non-zero element in $\rho^{(M)}(t)$ equals 1 and 0 for other positions.

*Example.* Let $p = 4$, $|\Gamma| = 4$ and $s_L = (5, 3, 1, 3, 7, 3, 3, 7, 1, 3, 5)$. Then it follows $\rho^{(d)}(s_L) = [0, 1, 0, 0, ..., 0]$.

Except $\rho^{(M)}(t)$, the definitions of other sub-encoding functions are the same as in the proof of Theorem 2 and are not repeated here.

Finally, define encoding function $\rho(x) : \mathcal{X} \to \mathbb{Q}^n$, where $n = 2|\Gamma| + \lceil \log_2 |Q| \rceil + |Q||\Gamma| + 10f + 11$ by:

$$\rho(q, s_L, s_R) = \rho^{(q)}(q) \oplus \rho_0^{(s)}(s_L) \oplus \rho_0^{(s)}(s_R) \oplus \rho^{(M)}(s_L) \oplus \rho^{(M)}(s_R) \oplus \rho^{(d)}(s_L) \oplus \rho^{(d)}(s_R) \oplus$$
$$\rho^{(r)}(s_{L,(1)}) \oplus \rho^{(r)}(s_{R,(1)}) \oplus \rho^{(h)}(h(|s_L|)) \oplus \rho^{(h)}(h(|s_R|)) \oplus \mathbf{0}, \tag{52}$$

where $\mathbf{0}$ is a zero vector of size $|Q||\Gamma| + 6f + 9$.

Let $\mathbf{v}(t) \in \mathbb{Q}^n$ denote the neurons' value of an RNN $\mathcal{T}_{W,\mathbf{b}}$ at time $t$. Initialize $\mathbf{v}(1) = \rho(x)$. It suffices to prove that there exists $\mathcal{T}_{W,\mathbf{b}}$ such that $\mathbf{v}(4) = \rho(\mathcal{P}_\mathcal{M}(x))$.

The RNN $\mathcal{T}_{W,\mathbf{b}}$ is constructed as follows. The $n$ neurons are classified into ten groups:

1. Stage neurons $\mathbf{c}(t) \in \{0, 1\}^2$, initialized with $\mathbf{c}(1) = \mathbf{0}$;

2. Entry neurons $\mathbf{e}(t) \in \{0, 1\}^{|Q||\Gamma| - 1}$, initialized with $\mathbf{e}(1) = \mathbf{0}$;

3. Temporary tape neurons $s_j'(t), s_j''(t) \in \mathbb{Q}$, initialized with $s_j'(1) = s_j''(1) = 0$, where $j \in \{L, R\}$;

4. Tape neurons $s_j(t) \in \mathbb{Q}$, initialized with $s_j(1) = \rho_0^{(s)}(s_j)$, where $j \in \{L, R\}$;

5. Readout neurons $\mathbf{r}_j(t) \in \{0, 1\}^{|\Gamma| - 1}$, initialized with $\mathbf{r}_j(1) = \rho^{(r)}(s_{j,(1)})$, where $j \in \{L, R\}$;

6. State neurons $\mathbf{q}(t) \in \{0, 1\}^{\lceil \log_2 |Q| \rceil}$, initialized with $\mathbf{q}(1) = \rho^{(q)}(q)$;

7. Guard neurons $g_j(t), g_j'(t), g_j''(t) \in \mathbb{Q}$, initialized with $g_j(1) = \rho^{(h)}(h(|s_j|))$ and $g_j'(1) = g_j''(1) = 0$, where $j \in \{L, R\}$;

8. Buffer neurons $\mathbf{o}_j(t), \boldsymbol{\beta}_j(t), \boldsymbol{\beta}_j'(t) \in \mathbb{Q}^f$, initialized with $\mathbf{o}_j(1) = \boldsymbol{\beta}_j(1) = \boldsymbol{\beta}_j'(1) = \mathbf{0}$, where $j \in \{L, R\}$;

9. Stack neurons $\mathbf{m}_j(t) \in \mathbb{Q}^f$, initialized with $\mathbf{m}_j(1) = \rho^{(M)}(s_j)$, where $j \in \{L, R\}$;

10. Pointer neurons $\boldsymbol{\pi}_j(t) \in \mathbb{Q}^f$, initialized with $\boldsymbol{\pi}_j(1) = \rho^{(d)}(s_j)$, where $j \in \{L, R\}$.

The general idea of the proof is to implement the growing memory module in Section 4 by an RNN as well and place all neurons inside the RNN. Neurons that are away from the position of the read/write head are stored in stack neurons $\mathbf{m}_j(t)$ instead of a separate stack in the growing memory module. Nonetheless, the storing mechanisms are similar, as each $p$ symbols are stored as a neuron using fractal encoding, except that we append $\mathbf{0}$ to the stack neurons to ensure that there are $f$ stack neurons. Therefore, the stack neurons can also be updated similarly to the growing memory module: if the tape neuron holds less than 1 symbol after the update, we pop the stack neuron (i.e. set the last non-zero neuron in $\mathbf{m}_j(t)$ to 0); If the tape neuron holds more than $p$ symbols after the update, we push the bottom $p$ symbols of the tape neuron to the stack neurons (i.e. set the first zero neuron in $\mathbf{m}_j(t)$ to the value to be pushed).

Except as otherwise stated, we use the same notation as in the proof of Theorem 2. Also, we denote $i_j^* = \lceil |s_j|/p - 1 \rceil$, which is the position of the last non-zero neuron in $\rho^{(M)}(s_j)$ (or $\mathbf{m}_j(1)$) for $j \in \{L, R\}$. Note that $\boldsymbol{\pi}_{j,i}(1)$ equals 1 if $i = i_j^*$, and 0 otherwise.

First, we have to update the formulas for buffer neurons to allow correct updates for the stack neurons:

*8. Buffer neurons.*

$$\mathbf{o}_{j,i}(t+1) = \sigma(\mathbf{m}_{j,i}(t) - (1 - \boldsymbol{\pi}_{j,i}(t)) - \mathbf{c}_1'(t) - \mathbf{c}_2'(t)) \tag{53}$$

$$\boldsymbol{\beta}_{L,i}(t+1) = \sigma(\hat{o}_L(t) - move_R(t) - (p+1)g_L(t) + 1 - (1 - \boldsymbol{\pi}_{L,i}(t)) - 2\mathbf{c}_2'(t) - 2\mathbf{c}_3'(t)), \tag{54}$$

$$\boldsymbol{\beta}_{L,i}'(t+1) = \sigma(s_L(t) - readSF(\mathbf{r}_L(t)) + writeSF(t) - move_L(t) - p + (p+1)g_L(t) - \\ (1 - \boldsymbol{\pi}_{L,i}(t)) - 2\mathbf{c}_2'(t) - 2\mathbf{c}_3'(t)), \tag{55}$$

$$\boldsymbol{\beta}_{R,i}(t+1) = \sigma(\hat{o}_R(t) - move_L(t) - (p+1)g_R(t) + 1 - (1 - \boldsymbol{\pi}_{R,i}(t)) - 2\mathbf{c}_2'(t) - 2\mathbf{c}_3'(t)), \tag{56}$$

$$\boldsymbol{\beta}_{R,i}'(t+1) = \sigma(s_R(t) - move_R(t) - p + (p+1)g_R(t) - (1 - \boldsymbol{\pi}_{R,i}(t)) - 2\mathbf{c}_2'(t) - 2\mathbf{c}_3'(t)), \tag{57}$$

where $j \in \{L, R\}$, $i \in \{1, 2, .., f\}$, $readSF$ and $writeSF$ are defined the same as in the proof of Theorem 1. Also, for $j \in \{L, R\}$, denote $\hat{o}_j(t) = \sum_{i=1}^f \mathbf{o}_{j,i}(t)$, $\hat{\beta}_j(t) = \sum_{i=1}^f \boldsymbol{\beta}_{j,i}(t)$, and $\hat{\beta}_j'(t) = \sum_{i=1}^f \boldsymbol{\beta}_{j,i}'(t)$.

The buffer neurons $\mathbf{o}_j(t)$ read the last non-zero neuron's value in $\mathbf{m}_j(t)$: $\mathbf{o}_{j,i}(2) = \mathbf{m}_{j,i}(1)$ if $i = i_j^*$ and 0 otherwise. Also, $\mathbf{o}_j(1) = \mathbf{o}_j(3) = \mathbf{o}_j(4) = \mathbf{0}$. It follows that $\hat{o}_j(2) = \mathbf{m}_{j,i_j^*}(1) = \rho_1^{(s)}(s_j)$ ($\rho_1^{(s)}$ is defined in the proof of Theorem 2), and $\hat{o}_j(1) = \hat{o}_j(3) = \hat{o}_j(4) = 0$. The purpose of $\hat{o}_j(t)$ is similar to the pop neuron $o_j(t)$ in the proof of Theorem 2, which reads the top neuron's value in the stack.

The equations for the remaining buffer neurons $\boldsymbol{\beta}_j(t)$ and $\boldsymbol{\beta}_j'(t)$ are almost the same as in the proof of Theorem 2, but with an additional term $-(1 - \boldsymbol{\pi}_{j,i}(t))$, which makes the neuron equals 0 if $i \neq i_j^*$. It follows that: i. $\boldsymbol{\beta}_{j,i}(3)$ equals to the value to be popped if popping is required and $i = i_j^*$, and 0 otherwise; ii. $\boldsymbol{\beta}_{j,i}'(3)$ equals to the value to be pushed if pushing is required and $i = i_j^*$, and 0 otherwise. Also, $\boldsymbol{\beta}_j(t) = \boldsymbol{\beta}_j'(t) = \mathbf{0}$ for $t \in \{1, 2, 4\}$. The dynamic of $\hat{\beta}_j(t)$ and $\hat{\beta}_j'(t)$ is similar but with the position requirement $i = i_j^*$ removed.

*9. Stack neurons.*

$$\mathbf{m}_{j,i}(t+1) = \sigma(\mathbf{m}_{j,i}(t) + \boldsymbol{\beta}_{j,i-1}'(t) - \boldsymbol{\beta}_{j,i}(t)), \tag{58}$$

where $j \in \{L, R\}$ and $i \in \{1, 2, .., f\}$. If the index is out of bound, then the term is set to 0; e.g. $\boldsymbol{\beta}_{j,0}'(t) = 0$. Note that $\mathbf{m}_j(1) = \mathbf{m}_j(2) = \mathbf{m}_j(3) = \rho^{(M)}(s_j)$. For $\mathbf{m}_j(4)$, there are three possible scenarios:

i. If no pushing or popping is required: $\boldsymbol{\beta}_j(3) = \boldsymbol{\beta}_j'(3) = \mathbf{0}$, hence $\mathbf{m}_j(4) = \mathbf{m}_j(1) = \rho^{(M)}(s_j) = \rho^{(M)}(s_j')$.

ii. If popping is required: $\rho^{(M)}(s_j')$ equals to $\rho^{(M)}(s_j)$ but with the last non-zero neuron set to 0. Since $\boldsymbol{\beta}_{j,i_j^*}(3)$ equals $\mathbf{m}_{j,i_j^*}(3)$ if popping is required, it follows that $\mathbf{m}_{j,i_j^*}(4) = 0$. Also, $\mathbf{m}_{j,i}(4) = \mathbf{m}_{j,i}(3)$ for $i \neq i_j^*$. This implies that $\mathbf{m}_j(4) = \rho^{(M)}(s_j')$.

iii. If pushing is required: $\rho^{(M)}(s_j')$ equals to $\rho^{(M)}(s_j)$ but with the first zero neuron set to the value to be pushed. Since $\boldsymbol{\beta}_{j,i_j^*}'(3)$ equals to the value to be pushed if pushing is required and $\mathbf{m}_{j,i_j^*+1}(4) = \boldsymbol{\beta}_{j,i_j^*}'(3)$, it follows that the first zero neuron in $\mathbf{m}_j(3)$ is set to the value to be pushed at $t = 4$. Also, $\mathbf{m}_{j,i}(4) = \mathbf{m}_{j,i}(3)$ for $i \neq i_j^* + 1$. This implies that $\mathbf{m}_j(4) = \rho^{(M)}(s_j')$.

Therefore, we have $\mathbf{m}_j(1) = \mathbf{m}_j(2) = \mathbf{m}_j(3) = \rho^{(M)}(s_j)$ and $\mathbf{m}_j(4) = \rho^{(M)}(s_j')$.

*10. Pointer neurons.*

$$\boldsymbol{\pi}_{j,i}(t+1) = \sigma(\boldsymbol{\pi}_{j,i}(t) + |2\Gamma|(-\boldsymbol{\beta}_{j,i}(t) - \boldsymbol{\beta}_{j,i}'(t) + \boldsymbol{\beta}_{j,i+1}(t) + \boldsymbol{\beta}_{j,i-1}'(t))), \tag{59}$$

where $j \in \{L, R\}$ and $i \in \{1, 2, .., f\}$. If the index is out of bound, then the term is set to 0. Note that $\boldsymbol{\pi}_j(1) = \boldsymbol{\pi}_j(2) = \boldsymbol{\pi}_j(3) = \rho^{(d)}(s_j)$ . For $\boldsymbol{\pi}_j(4)$, there are three possible scenarios:

i. If no pushing or popping is required: $\boldsymbol{\beta}_j(3) = \boldsymbol{\beta}'_j(3) = \mathbf{0}$, hence $\boldsymbol{\pi}_j(4) = \boldsymbol{\pi}_j(1) = \rho^{(d)}(s_j) = \rho^{(d)}(s'_j)$.

ii. If popping is required: $\rho_i^{(d)}(s'_j)$ equals to 1 if $i = i^*_j - 1$, and 0 otherwise. Since $\boldsymbol{\beta}_{j,i^*_j}(3)$ is larger than $1/|2\Gamma|$ if popping is required, it follows that $\boldsymbol{\pi}_{j,i^*_j}(4) = 0$ and $\boldsymbol{\pi}_{j,i^*_j-1}(4) = 1$. Also, $\boldsymbol{\pi}_{j,i}(4) = \boldsymbol{\pi}_{j,i}(3) = 0$ for $i \notin \{i^*_j - 1, i^*_j\}$. This implies that $\boldsymbol{\pi}_j(4) = \rho^{(d)}(s'_j)$.

iii. If pushing is required: $\rho_i^{(d)}(s'_j)$ equals to 1 if $i = i^*_j + 1$, and 0 otherwise. Since $\boldsymbol{\beta}'_{j,i^*_j}(3)$ is larger than $1/|2\Gamma|$ if pushing is required, it follows that $\boldsymbol{\pi}_{j,i^*_j}(4) = 0$ and $\boldsymbol{\pi}_{j,i^*_j+1}(4) = 1$. Also, $\boldsymbol{\pi}_{j,i}(4) = \boldsymbol{\pi}_{j,i}(3) = 0$ for $i \notin \{i^*_j, i^*_j + 1\}$. This implies that $\boldsymbol{\pi}_j(4) = \rho^{(d)}(s'_j)$.

Therefore, we have $\boldsymbol{\pi}_j(1) = \boldsymbol{\pi}_j(2) = \boldsymbol{\pi}_j(3) = \rho^{(d)}(s_j)$ and $\boldsymbol{\pi}_j(4) = \rho^{(d)}(s'_j)$. Also note that if $\boldsymbol{\pi}_j(4) = \mathbf{0}$, it implies that the Turing Machine has reached the end of the tape and has to halt.

The update equations for the remaining neurons from 1. to 7. are the same as in the proof of Theorem 2, except that $\beta_j(t)$ and $\beta'_j(t)$ in the update equations of tape neurons and readout neurons are replaced with $\hat{\beta}_j(t)$ and $\hat{\beta}'_j(t)$ respectively. Together, $\mathbf{v}(4) = \rho(\mathcal{P}_\mathcal{M}(x))$. This completes the proof. $\qquad\square$

# D    Notation Table

| Symbol | Description | Defined in |
|---|---|---|
| | **Turing Machine** | |
| $\mathcal{M}$ | Turing Machine of a Turing Machine | Section 2 |
| $Q$ | Finite set of state | Section 2 |
| $F$ | Finite set of final state | Section 2 |
| $\Sigma$ | Finite set of input symbols | Section 2 |
| $\Gamma$ | Finite set of tape symbols | Section 2 |
| $\delta$ | Transition rule | Section 2 |
| $q_0$ | The initial starting state | Section 2 |
| $q$ | The current state | Section 2 |
| $\sharp$ | The blank symbol | Section 2 |
| $s_L$ | The string of symbols under and left to the read/write head | Section 2 |
| $s_R$ | The string of symbols right to the read/write head | Section 2 |
| $\mathcal{X}$ | Sets of all possible instantaneous description | Section 2 |
| $\mathcal{P}_{\mathcal{M}}$ | Complete dynamic map of $\mathcal{M}$ (or equivalently, one transition of $\mathcal{M}$) | Section 2 |
| $\mathcal{P}_{\mathcal{M}}^*$ | Partial input-output function of $\mathcal{M}$ (or equivalently, the input-output function defined by $\mathcal{M}$) | Section 2 |
| $U_{6,4}$ | A particular Universal Turing Machine with 6 states and 4 symbols | Section 3 |
| | **RNN** | |
| $\sigma$ | Saturated-linear function | Equation (1) |
| $W, \mathbf{b}$ | Parameters of the RNN | Section 2 |
| $x_i(t)$ | The value of neuron $i$ at time $t \in \{1, 2, ...\}$ | Section 2 |
| $t$ | Time step of RNN | Section 2 |
| $\mathcal{T}_{W,\mathbf{b}}$ | Mapping defined by RNN with parameters $W, \mathbf{b}$ (or equivalently, one step of RNN) | Section 2 |
| $\mathcal{T}_{W,\mathbf{b}}^3$ | Apply $\mathcal{T}_{W,\mathbf{b}}$ three times | Section 2 |
| $\mathcal{T}_{W,\mathbf{b}}^*$ | Apply $\mathcal{T}_{W,\mathbf{b}}$ repeatedly until the decoded state reaching the final state | Section 3 |
| $u(t)$ | The value of push neuron in growing memory modules | Section 3 |
| $o(t)$ | The value of pop neuron in growing memory modules | Section 3 |
| $g_L(t), g_R(t)$ | The value of left-guard and right-guard neuron in growing memory modules | Section 3 |
| | **Encoding Function** | |
| $\rho^{(q)}$ | Encoding function for the state | Section 3 |
| $\rho^{(s)}$ | Encoding function for the tape (two versions) | Equation (3), (8) |
| $\rho^{(r)}$ | Encoding function for the top symbol in tapes | Equation (4) |
| $\rho^{(h)}$ | Encoding function for the number of symbols in tapes | Equation (9) |
| $\rho^{(M)}$ | Encoding function for the stack | Section 4 |
| $\rho$ | Encoding function (two versions) | Equation (5), (10) |
| $\rho^{-1}$ | Decoding function (inverse of the encoding function) | Equation (5), (10) |
| $h$ | Counting function, which counts the number of symbols residing in the RNN | Section 4 |
| | **Miscellaneous** | |
| $a_{(i)}$ | $i^{\text{th}}$ symbol of string $a$ | Section 2 |
| $a_{(i:j)}$ | Sub-string $a_{(i)}a_{(i+1)}...a_{(j)}$ of string $a$ | Section 2 |
| $\mathbb{Q}$ | The set of rational number | Section 2 |
| $\mathbf{x} \oplus \mathbf{y}$ | Concatenation of two vectors $\mathbf{x}$ and $\mathbf{y}$ | Section 2 |
| $A^*$ | All possible strings formed by elements from set $A$ | |

Table 1: Notation used in the main paper.