# OpenReview forum: "Turing Completeness of Bounded-Precision Recurrent Neural Networks"
_NeurIPS.cc/2021/Conference — NeurIPS 2021 Poster_

### Official Review · Reviewer_S4CZ · 2021-07-16

**Rating:** 7
**Confidence:** 2

**Summary:**

This theoretical work shows the existence of un-bounded precision RNNs for simulating turing machines, with growing/shrinking memory module. Although the paper seems quite interesting and well-written, I am afraid I don't have the right background to make an accurate judgment on the novelty and potential impact of the work. I share few thoughts below, similarly due to lack-of-knowledge and limited reviewing time I was not able to check the proofs.

**Limitations And Societal Impact:**

This is a theoretical work, so no societal impact I can think of. Authors also discuss limitations at the end, which I appreciated.

**Main Review:**

# Strengths
- Most of the results seems novel and paper is very well-written.

# Questions/Possible-Limitations
- Given proof shows the existence of bounded/non-bounded precision RNNs with particular size (for simulating turing machines) and these networks are not-trainable (due to non-differentiable parts). As mentioned in the conclusion this is a limitation and I wonder (apart from inherit value of it) what other values the proposed architecture might bring to the field (practical/theoretical). This part is not clear to me and I think it would be nice to discuss this in the rebuttal. Can we train similar architectures if we make growing differentiable? What would that enable that is not already enabled?
- I wonder whether copying experiments (like in NTM paper) is appropriate here. It would be nice to have a working implementation of proposed RNN.

## Minor
- Without loss of generosity, -> generality.
- "Implementing Neural Turing Machines" https://arxiv.org/abs/1807.08518 Might be a relevant work.
- "https://www.nature.com/articles/nature20101" Same with this.

**Time Spent Reviewing:**

3

---

> ### Author Response · Authors · 2021-08-10
> **Response to R4**
>
> We thank the reviewer for the constructive comments and detailed feedback. We appreciate the positive feedback very much. We would also like to stress the theoretical importance of Theorem 6, a novel theorem on the Turing Completeness of bounded-precision RNNs, given the ubiquity of bounded-precision RNNs in machine learning. Below are our responses to the comments:
>
> 1. What other values the proposed architecture might bring to the field (practical/theoretical).
>
> First, the proposed architecture is able to simulate a Turing machine efficiently with only bounded-precision neurons while maintaining a constant time complexity with respect to the size of memory. Second, the proposed architecture has theoretical significance: its Turing completeness allows us to establish Theorem 6, a novel theorem on bounded-precision RNNs that is important by itself, and also the Turing completeness of stack RNNs that have been recently used for gaining practical capabilities. Third, as the proposed simulation explicitly constructs a bounded-precision RNN to simulate a Turing machine, it offers a new method of incorporating prior knowledge in the network: given a Turing machine, one can initialize the weights of an RNN (both stack RNN or bounded-precision RNN) to simulate the Turing machine and fine-tune the weights using learning. And last but not least, by proving Turing completeness, we clarify that neural network representations are able to embed algorithms, and this opens the door to efforts of pre- or post-training inclusion of information as well as learning discrete algorithms.
>
> 2. Can we train similar architectures if we make growing differentiable? What would that enable that is not already enabled?
>
> It is possible to train the growing memory module using REINFORCE, similar to [1]. We think that this would enable handling tasks that require large memory storage, such as the copying experiments as suggested by the reviewer, and we intend to work on it in follow-up work.
>
> [1] Zaremba, W., & Sutskever, I. (2015). Reinforcement learning neural turing machines-revised. arXiv preprint arXiv:1505.00521.

---

### Official Review · Reviewer_jTHy · 2021-07-16

**Rating:** 9
**Confidence:** 3

**Summary:**

This work proposes a dynamic-growing memory module for RNNs, which serves to simulate Turing machines of bounded precision.
First, the authors prove how to encode a Turing machine to an unbounded precision RNN. The encoding uses fractal encoding for symbols, like previous work, and it replaces each step of the Turing machine as 3 steps in the RNN. Leaning in the work of [14], they prove that there is a 40-neuron unbounded precision RNN to simulate any Turing machine (i.e., showing Turing-completeness). Next, they consider the previous unbounded-precision definitions with the memory module to obtain a bounded-precision RNN.The memory module is a stack of neurons, that requires two neurons in the RNN to control it: push and pop. Then it is proven that the proposed bounded-precision RNN utilizes 2 of these stack to simulate a Turing machine. Further, each stack is divided in groups of $p$ neurons, such that the most active neurons are in the RNN (near to each side of the memory head in the Turing machine). Leaning again in [14], they can show that there is a 54-neuron with $p$-precision RNN with 2 growing memory module, that can simulate any Turing machine. This proves Turing-completeness of the precision-bounded RNN. Finally, they remove the growing memory module and simulate the memory within the neurons of an RNN, showing that an infinite number of bounded-precision RNN can simulate any Turing machine. the work finishes with an interesting discussion of the results and limitations.


**Ethical Concerns:**

None to the best of my knowledge.


**Limitations And Societal Impact:**

Limitations and societal impact have been addressed to the best of my understanding.


**Main Review:**

The theoretical results presented in this work are novel to the best of my knowledge. The ideas to build the mappings are simple and lean, citing related work as needed. The idea of using stacks with RNNs is not novel, the authors may want to have a look at the works [1],[2],[3] that apply it in practice (and require a differentiable stack).

The paper is clear and proofs look sound. The claims are properly supported and discussed, facilitating the understanding of the work. The flow of the text read smoothly (despite the number of symbols). On one hand, the results improve over the existing state-of-the-art in terms of the number of neurons for unbounded precision. On the other hand, shows that bounded-precision RNNs with memory are Turing-complete. Section 5 and the final discussion are a very interesting ending to this paper. A little more detail for Figure 2 in the main paper will be highly appreciated by this reader.

Significance: Are the results important? Are others (researchers or practitioners) likely to use the ideas or build on them? Does the submission address a difficult task in a better way than previous work? Does it advance the state of the art in a demonstrable way? Does it provide unique data, unique conclusions about existing data, or a unique theoretical or experimental approach?

This work is clearly very important to the NeurIPS community and the deep learning community in particular. This work does not only make solid theoretical contributions where needed, but also emphasizes the importance of memory augmentation in RNNs.

It was very pleasant to read this paper and the nice results. Thank you!


Minor:
* Line 112: "without loss of generosity" --> "without loss of generality"

References

[1] Mali et al. "Recognizing Long Grammatical Sequences Using Recurrent Networks Augmented With An External Differentiable Stack", 2020.

[2] Mali et al. "The Neural State Pushdown Automata", IEEE Transactions on Artificial Intelligence, 2020.

[3] Suzgin et al. "Memory-Augmented Recurrent Neural Networks Can Learn Generalized Dyck Languages", 2019.

**Time Spent Reviewing:**

4

---

> ### Author Response · Authors · 2021-08-10
> **Response to R3**
>
> It was pleasant that the reviewer recognizes the theoretical contribution of our paper. The summary is complete, and the comments are constructive. We appreciate the positive feedback very much.
>
> We will update the caption of Figure 2 to make it clearer. The figure shows the RNN that we construct to simulate a Turing machine, with neurons separated into different groups based on their role in the simulation.

---

### Official Review · Reviewer_9x9n · 2021-07-17

**Rating:** 6
**Confidence:** 3

**Summary:**

The paper presents 3 main theoretical results:
- An RNN with 40 unbounded-precision neurons is Turing-complete.
- An RNN with 54 bounded-precision neurons and two stacks is Turing-complete.
- An RNN with a finite number of bounded-precision neurons and no stacks can simulate a Turing machine with a bounded tape, where the maximum tape length is related to the number of RNN neurons.

**Limitations And Societal Impact:**

See the main review for limitations.

**Main Review:**

The paper has a limited practical significance.

About unbounded-precision neurons:
- Unbounded-precision is not practical.

About the RNN with bounded-precision neurons and two stacks:
- It is well known that a finite-state machine with two stacks is Turing-complete.
And an RNN with bounded-precision is a finite-state machine.
- The proposed stacks are not differentiable.
- The original Neural Turing Machine was already Turing-complete, if using an unbounded tape.
The location-based addressing allows to shift the tape. By shifting a one-hot attention pattern, only the attended part of the tape needs to be accessed. This was practically used by Spatial Transformer Networks (NeurIPS 2015) and later works.



**Time Spent Reviewing:**

2

---

> ### Author Response · Authors · 2021-08-10
> **Response to R2**
>
> We thank the reviewer for the constructive comments and detailed feedback. We would like to emphasize that the main goal of our paper is to establish fundamental theories on (i) unbounded-precision RNNs, (ii) bounded-precision RNNs, and (iii) stack RNNs (or RNNs with growing memory modules) regarding Turing completeness; we do not propose new variants of RNNs that achieve state-of-the-art empirical performance. The importance in proving Turing completeness is the ability to embed any computer algorithm in the network; without this, it is not clear that networks are able to embed such algorithms and may consider to be inferior. By proving Turing completeness, we clarify that neural network representations are able to embed algorithms, and this opens the door to efforts of pre- or post-training inclusion of information as well as learning discrete algorithms. The main contribution of our paper rests on the new theorems regarding (ii) and (iii), as stated by the title of the paper under consideration.
>
> To facilitate discussion, we will provide some additional background of theorems relating RNNs and Turing completeness:
>
> 1.	First work on differentiable neural networks that simulate Universal Turing machines was published in 1996 by [1]. However, it assumes an unbounded precision of neurons.
> 2.	[2,3] proposed external memories to strengthen a finite automata controller. However, the memory is bounded in size due to the manner of accessing it, and hence these models are not Turing-complete.
> 3.	[4] suggested a dynamically growing memory, but the time complexity is linear to the size of the memory since all units in the memory have to be updated. In this way, it is similar to theoretical neural network designs from the 80’s.
> 4.	[5-7] proposed variants of trainable stack RNNs that have unbounded memory, and the time complexity is constant to the size of the memory.
> 5.	However, it has not been shown that [2-7] are Turing-complete when using bounded-precision neurons, or how these models can be used to simulate a Turing machine. To prove it, we provide the detailed construct of a memory-augmented RNN that is Turing-complete and explicitly derive the formulas and number of neurons required to simulate a Turing machine. The proof of Turing-completeness of our model also establishes the Turing-completeness of most variants of stack RNNs [4-7] (see our response 2. to Reviewer 1 for details).
> 6.	There is no prior work on Turing completeness of bounded-precision RNNs (without any memory).
>
> Below are our responses to the comments:
>
> 1. Unbounded-precision is not practical.
>
> This is acknowledged by the paper and is the major motivation of studying bounded-precision RNNs and stack RNNs in our papers. Nonetheless, unbounded-precision RNNs have important theoretical values: [8,9] established the Turing-completeness of unbounded-precision RNNs in the 1990s and has had an immense influence on works relating to RNNs, including LSTMs and NTMs, over the past two decades. This work is an extension of [8,9] which removes the assumptions of unbounded precision. The attractiveness of the unbounded-precision models in [8,9] is that the computation time is constant with respect to the size of the memory, similar to a Turing machine. This is not shared by most models such as [2-4]. Here we design and prove how to use bounded-precision neurons to simulate a Turing machine while keeping the constant simulation time.
>
> 2. It is well known that a finite-state machine with two stacks is Turing-complete. And an RNN with bounded-precision is a finite-state machine.
>
> The reviewer suggests: (i) For each finite-state machine (FSMs), there is a bounded-precision RNN that simulates it; (ii) An FSM with two stacks can simulate any Turing machines. Therefore, a bounded-precision RNN with two stacks can simulate any Turing machines. The implied statement is that Theorems 4 and 5 are not new results. (We can only guess the intended conclusion of the reviewer since it is not explicitly stated. Please correct us if we are wrong. Also, "a bounded-precision RNN is an FSM" is not equivalent to our (i) interpretation, but the former statement does not entail the conclusion, so we replaced it with the latter statement.)
>
> Both (i) and (ii) have been proven [10]. However, the conclusion does not follow from them. There is a missing gap between (i) and (ii) and the conclusions of this paper:  (iii)  A bounded-precision RNN with two stacks can simulate an FSM with two stacks. Though it may seem possible to derive (iii) based on the proof of (i) in [10], the proof is far from trivial: A rigorous definition of stacks and the correspondence between pushing and popping operations in the two systems (RNNs and FSMs) has to be established (including, e.g., what are the equations for push and pop neurons?).  Moreover, other technical issues exist. For example, it takes at least two computation steps to compute the correct push/pop action using the proof in [10] (one for detecting coincidence for state and input, another one for mapping combination of state and input to push / pop action), leading to a one-time-step lag in the reading from the stack that the RNN simulation needs to take care of.
>
> 3. The proposed stacks are not differentiable.
>
> As discussed in the paper and pointed out by the reviewer, the growing memory module is non-differentiable, and so it cannot be trained directly by SGD. We acknowledge this observation. After the first Turing equivalent differentiable network [1], much work [5-7] has been done to train similar stack RNNs with some success. There are also methods to train networks with non-differentiable operations, which were introduced in relation to NTM [11] that can be applied directly to our model, as well as biological inspired local learning methods (e.g., STDP and variants). Nonetheless, we state that the major value of the proposed growing memory module lies in its theoretical significance – its Turing completeness allows us to establish Theorem 6, a novel theorem on bounded-precision RNNs, and also the Turing completeness of similar stack RNNs. As such, it is outside the scope of this paper to incorporate an empirical evaluation. To repeat - the focus of the paper is the theoretical capability and Turing-completeness of various forms of RNNs.
>
> 4. The original Neural Turing Machine was already Turing-complete, if using an unbounded tape
>
> It is not the goal of our paper to propose new variants of RNNs that have state-of-the-art empirical performance. We propose the growing memory module due to its theoretical significance: it leads to new theorems on the Turing completeness of bounded-precision RNNs (Theorem 6) and allows us to establish the Turing completeness of stack RNNs.
>
> In addition, to our knowledge, it has not been proven that a bounded-precision NTM with unbounded memory is Turing-complete. However, this can be proven true by showing that an NTM with unbounded memory can simulate any RNNs with growing memory modules and Theorem 4. This again demonstrates the theoretical significance of Theorem 4 and the proposed growing memory modules.
>
> Finally, we would like to point out that given the ubiquity of bounded-precision RNNs, Theorem 6, a novel theorem on the Turing Completeness of bounded-precision RNNs, is arguably the most important theorem in the paper RNNs and thus the title of the paper. The significance of the theorem should not be easily overlooked despite being in the last section of the paper.
>
> [1] Kilian, J., & Siegelmann, H. T. (1996). The dynamic universality of sigmoidal neural networks. Information and computation, 128(1), 48-56.
>
> [2] Graves, A., Wayne, G., & Danihelka, I. (2014). Neural turing machines. arXiv preprint arXiv:1410.5401.
>
> [3] Graves, A., Wayne, G., Reynolds, M., Harley, T., Danihelka, I., Grabska-Barwińska, A., ... & Hassabis, D. (2016). Hybrid computing using a neural network with dynamic external memory. Nature, 538(7626), 471-476.
>
> [4] Joulin, Armand, and Tomas Mikolov. "Inferring algorithmic patterns with stack-augmented recurrent nets." Advances in neural information processing systems 28 (2015): 190-198.
>
> [5] Grefenstette, E., Hermann, K. M., Suleyman, M., & Blunsom, P. (2015). Learning to transduce with unbounded memory. Advances in neural information processing systems, 28, 1828-1836.
>
> [6] Sun, G. Z., Giles, C. L., Chen, H. H., & Lee, Y. C. (2017). The neural network pushdown automaton: Model, stack and learning simulations. arXiv preprint arXiv:1711.05738.
>
> [7] Mali, A., Ororbia, A., Kifer, D., & Giles, C. L. (2020). Recognizing long grammatical sequences using recurrent networks augmented with an external differentiable stack. arXiv preprint arXiv:2004.07623.
>
> [8] Siegelmann, H. T., & Sontag, E. D. (1995). On the computational power of neural nets. Journal of computer and system sciences, 50(1), 132-150.
>
> [9] Siegelmann, H. T. (1995). Computation beyond the Turing limit. Science, 268(5210), 545-548.
>
> [10] Minsky, M. L. (1967). Neural Network. In Computation: Finite and Infinite Machines (pp. 144-163). Englewood Cliffs: Prentice-Hall.
>
> [11] Zaremba, W., & Sutskever, I. (2015). Reinforcement learning neural turing machines-revised. arXiv preprint arXiv:1505.00521.

---

> > ### Comment · Reviewer_9x9n · 2021-08-18
> > **Reply**
> >
> > Thank you for your detailed response.
> >
> > I agree with you in almost everything.
> > I do not agree with the background point 2:
> > "[2,3] proposed external memories to strengthen a finite automata controller. However, the memory is bounded in size due to the manner of accessing it, and hence these models are not Turing-complete."
> >
> > As I mentioned in my review, the NTM memory size does not have to be bounded. The Spatial Transformer implemented the location based addressing with O(1) time complexity, independent of the memory size.
> >
> > You care about precise proofs. I acknowledge your hard work. Please, care also about precise historical background.

---

> > > ### Author Response · Authors · 2021-08-19
> > > **Response to R2**
> > >
> > > We thank the reviewer for the response and the acknowledgment. We acknowledge that some related work is not adequately addressed in the paper, and we will add the historical background in the revised version of the paper.
> > >
> > > On the unbounded memory of NTMs – We agree that it is possible to use unbounded memory on NTM if (i) the content-based addressing is removed (since the key vector has to be compared to all memory) and (ii) the attention weighting vector is initialized to be sparse (e.g., one-hot vector as suggested by the reviewer, so time complexity of reading or writing is O(1)). The resulting NTM shares some similarities with a stack RNN, which gives an interesting relation between NTMs and stack RNNs. We thank the reviewer for pointing out this, and we will add this discussion to the paper.

---

### Official Review · Reviewer_fUf1 · 2021-07-17

**Rating:** 3
**Confidence:** 5

**Summary:**

This paper analyses the Turing completeness of recurrent neural networks, and proposes a memory-augmented RNN architecture reminiscent of existing stack RNNs.

Specifically:

First, a construction for simulating any Turing machine with an unbounded-precision RNN is presented, in which effectively the left and right sides of the Turing machine's tape (with respect to its head) are encoded as two 'stacks' in the RNN. This construction is analysed and requires less simulation time than that of Siegelmann and Sonntag [1] (3T instead of 4T+O(used tape length), where T is computation time of the simulated Turing machine). Additionally the authors analyse the exact precision of the neurons necessary (as function of the simulated Turing machine's computation length) for the simulation (as opposed to previous works, which simply note a requirement for infinite precision).

Second, noting that it is unavoidable that the precision must grow when simulating longer Turing machine computations (to maintain the contents of the tape), the authors propose augmenting the RNN with a dynamic memory module, such that the neurons themselves may have constant, bounded, precision. They theoretically analyse their proposed architecture and show how a Turing machine may be simulated in it.



**Limitations And Societal Impact:**

none relevant

**Main Review:**

The construction of this dynamic memory module is  reminiscent of recent work on "stack RNNs" [2-4]. Much like the model proposed in this paper, stack RNNs push and pop to an external 'memory module' (generally referred to as a 'stack'), which allows them to simulate pushdown automata (although most of these models do not discuss 2 stacks as used here, their constructions can certainly be generalised to multiple stacks, and so it is conceivable that once sufficient success is achieved with one stack then there would be a shift to 2 stacks or more). Much of the challenge with these stack-RNNs is making this stack differentiable, so that the model can 'learn' whether it should be pushing or popping from its stack during training (i.e., the push and pop operations need to be differentiable). Multiple different methods are considered to attain this quality (see partial list of works below). In this paper it does not seem that this challenge has been addressed, and indeed the current push and pop operations appear to be happening discretely, i.e., I do not think it is possible to train the current architecture with SGD.

Another thing I would like to note is that it is not generally convincing to propose a new architecture without some practical evaluation. While the theoretical power of this architecture is shown in this work (a construction for simulating any Turing machine), its ability to learn any language, formal or natural, unfortunately has not been evaluated. I suspect that this is also why the challenge of making differentiable push and pop operations has gone unnoticed in this work.

To conclude:
1. In itself the construction of a Turing machine simulation in RNNs is not new (e.g., it has already been done in [1]), but I appreciate the analysis showing that this construction is faster than previous constructions, and moreover the analysis of necessary precision per Turing machine and input, which (if I remember correctly) is not present in previous such works. However, I do not see this result as sufficient for accepting this paper alone: I believe the main interest of the community in this line of results is more the fact that RNNs *can* simulate Turing machines, rather than how efficiently they can do so. (Maybe if the improved construction was significantly more efficient, that would be nice, but this is only a constant-factor improvement: from 5T (we can say that the tape will never be longer than T) to 3T).
2. I appreciate the attempt to create a memory augmented network. However at this point I would only want to accept new architectures after they have been evaluated, and this one has not (after all, if we wanted only to *encode* Turing machines in them, then we can simply use those Turing machines directly instead). Moreover I suspect that if we do come to train this architecture, we will hit many obstacles (see e.g. discreteness of push and pop operations, above).

If the authors pursue making this architecture differentiable, I strongly suggest they read more about stack RNNs (I am not an expert on these, so I give only a partial list), where I think they will find a lot of inspiration. Either way, I ask that they evaluate their architecture, and moreover compare it to existing stack RNNs and at the very least vanilla RNNs (including variants such as GRUs and LSTMs), before they resubmit.



references:
[1] On the computational power of neural nets (Siegelmann and Sonntag)

[2] Inferring Algorithmic Patterns with Stack-Augmented Recurrent Nets (Joulin and Mikolov)

[3] The Neural Network Pushdown Automaton: Model, Stack and Learning Simulations (Sun, Giles, Chen, Lee)

[4] Learning to Transduce with Unbounded Memory (Grefenstette, Hermann, Suleyman, Blunsom)


**Time Spent Reviewing:**

7

---

> ### Author Response · Authors · 2021-08-10
> **Response to R1:**
>
> We thank the reviewer for the constructive comments and detailed feedback. We would like to emphasize that the main goal of our paper is to establish fundamental theories on (i) unbounded-precision RNNs, (ii) bounded-precision RNNs, and (iii) stack RNNs (or RNNs with growing memory modules) regarding Turing completeness; we do not propose new variants of RNNs that achieve state-of-the-art empirical performance. The importance in proving Turing completeness is the ability to embed any computer algorithm in the network; without this, it is not clear that networks are able to embed such algorithms and may consider to be inferior. By proving Turing completeness, we clarify that neural network representations are able to embed algorithms, and this opens the door to efforts of pre- or post-training inclusion of information as well as learning discrete algorithms. The main contribution of our paper rests on the new theorems regarding (ii) and (iii), as stated by the title of the paper under consideration.
>
> To facilitate discussion, we will provide some additional background of theorems relating RNNs and Turing completeness:
>
> 1.	First work on differentiable neural networks that simulate Universal Turing machines was published in 1996 by [1]. However, it assumes an unbounded precision of neurons.
> 2.	[2,3] proposed external memories to strengthen a finite automata controller. However, the memory is bounded in size due to the manner of accessing it, and hence these models are not Turing-complete.
> 3.	[4] suggested a dynamically growing memory, but the time complexity is linear to the size of the memory since all units in the memory have to be updated. In this way, it is similar to theoretical neural network designs from the 80’s.
> 4.	[5-7] proposed variants of trainable stack RNNs that have unbounded memory, and the time complexity is constant to the size of the memory.
> 5.	However, it has not been shown that [2-7] are Turing-complete when using bounded-precision neurons, or how these models can be used to simulate a Turing machine. To prove it, we provide the detailed construct of a memory-augmented RNN that is Turing-complete and explicitly derive the formulas and number of neurons required to simulate a Turing machine. The proof of Turing-completeness of our model also establishes the Turing-completeness of most variants of stack RNNs [4-7] (see response 2. below for details).
> 6.	There is no prior work on Turing completeness of bounded-precision RNNs (without any memory).
>
> Below are our responses to the comments:
>
> 1. In itself the construction of a Turing machine simulation in RNNs is not new.
>
> Previous complete proofs that establish Turing-completeness of RNNs are limited to the case of unbounded-precision RNNs. Hence, our theorems about the universal capability of both bounded-precision RNNs and stack RNNs in terms of simulating Turing machines are novel. Given the popularity of recurrent networks and the somewhat unsupported statements about their capabilities, we believe these new theorems constitute an important foundation to the field. See in Reviewer 3: “This work does not only make solid theoretical contributions where needed, but also emphasizes the importance of memory augmentation in RNNs”.
>
> 2. Relationship between stack RNNs and the proposed RNNs with growing memory modules.
>
> A closer inspection of our definition of growing memory modules (Definition 3) reveals that it is a basic foundational form of stack RNNs, which can easily be shown to be isomorphic to various stack RNNs proposed in the past [5-7]. For example, an RNN with neural stacks [5] can simulate any RNNs with growing memory modules: for pushing, set $v_t = z_t$ and $d_t = 1[z_t > 0]$; for popping, set $u_t = 1[k_t=0]$ (the L.H.S. correspond to units in the neural stack and R.H.S. correspond to units in the growing memory module). Therefore, one can view our proposed model as a generic version of stack RNNs.
>
> It follows from Theorem 4 that any network that is isomorphic to our proposed RNNs with growing memory modules is Turing-complete. Thus, our theorem also extends to most stack RNNs with an unbounded memory and establishes theoretical motivation for many stack RNNs in the literature.
>
> We acknowledge that we did not clarify this relationship between the proposed growing memory modules and stack RNNs, which is a critical point. We will include the above discussion in the revised version of the paper.
>
> 3. The proposed growing memory module is non-differentiable and lack of empirical evaluation.
>
> As discussed in the paper and pointed out by the reviewer, the growing memory module is non-differentiable, and so it cannot be trained directly by SGD. We acknowledge this observation. After the first Turing equivalent differentiable network [1], much work [5-7] has been done to train similar stack RNNs with some success. There are also methods to train networks with non-differentiable operations, which were introduced in relation to NTM [8] that can be applied directly to our model, as well as biological inspired local learning methods (e.g., STDP and variants). Nonetheless, we state that the major value of the proposed growing memory module lies in its theoretical significance – its Turing completeness allows us to establish Theorem 6, a novel theorem on bounded-precision RNNs, and also the Turing completeness of similar stack RNNs. As such, it is outside the scope of this paper to incorporate an empirical evaluation. To repeat - the focus of the paper is the theoretical capability and Turing-completeness of various forms of RNNs.
>
> [1] Kilian, J., & Siegelmann, H. T. (1996). The dynamic universality of sigmoidal neural networks. Information and computation, 128(1), 48-56.
>
> [2] Graves, A., Wayne, G., & Danihelka, I. (2014). Neural turing machines. arXiv preprint arXiv:1410.5401.
>
> [3] Graves, A., Wayne, G., Reynolds, M., Harley, T., Danihelka, I., Grabska-Barwińska, A., ... & Hassabis, D. (2016). Hybrid computing using a neural network with dynamic external memory. Nature, 538(7626), 471-476.
>
> [4] Joulin, Armand, and Tomas Mikolov. "Inferring algorithmic patterns with stack-augmented recurrent nets." Advances in neural information processing systems 28 (2015): 190-198.
>
> [5] Grefenstette, E., Hermann, K. M., Suleyman, M., & Blunsom, P. (2015). Learning to transduce with unbounded memory. Advances in neural information processing systems, 28, 1828-1836.
>
> [6] Sun, G. Z., Giles, C. L., Chen, H. H., & Lee, Y. C. (2017). The neural network pushdown automaton: Model, stack and learning simulations. arXiv preprint arXiv:1711.05738.
>
> [7] Mali, A., Ororbia, A., Kifer, D., & Giles, C. L. (2020). Recognizing long grammatical sequences using recurrent networks augmented with an external differentiable stack. arXiv preprint arXiv:2004.07623.
>
> [8] Zaremba, W., & Sutskever, I. (2015). Reinforcement learning neural turing machines-revised. arXiv preprint arXiv:1505.00521.

---

> > ### Comment · Reviewer_fUf1 · 2021-08-31
> > **quick question**
> >
> > Dear Authors, thank you for noting that your growing memory module can emulate a stack RNN. Sorry to ask at short notice, but would you mind clarifying the following: can a stack RNN also emulate the growing memory module (and if so, could you explain how)? (I.e., does the inverse relationship hold?) This is important to understand the relationship between the two.
> > Thank you!

---

> > > ### Author Response · Authors · 2021-09-01
> > > **Response to R1**
> > >
> > > Thank you for the question. As the proposed growing memory module is a basic foundational form of stack RNNs, most stack RNNs are isomorphic to our proposed growing memory modules, i.e., (i) stack RNNs can simulate any growing memory modules, and (ii) growing memory modules can simulate any stack RNNs. We will use the stack RNN proposed in [1] as an example:
> > >
> > > Proof of (i): For pushing operation, set $v_t$ in the stack RNN to $z_t$ in the growing memory module, and $d_t$ in the stack RNN to $1[z_t > 0]$ in the growing memory module, where $1[\cdot]$ is the indicator function. For popping operation, set $u_t$ in the stack RNN to $1[k_t = 0]$ in the growing memory module. One step of the stack RNN can then simulate one step of the given growing memory module, as the values of the stacks and the neurons in the RNNs are the same.
> > >
> > > Proof of (ii): It follows from the fact that growing memory modules are Turing-complete and stack RNNs are Turing computable.
> > >
> > > Therefore, our proof of the Turing completeness of (bounded-precision) growing memory modules extends to other (bounded-precision) stack RNNs. We agree that it is important to understand the relationship between the two, and we thank the reviewer for pointing out this. We will add this discussion to the paper.
> > >
> > > [1] Grefenstette, E., Hermann, K. M., Suleyman, M., & Blunsom, P. (2015). Learning to transduce with unbounded memory. Advances in neural information processing systems, 28, 1828-1836.

---

> > > > ### Comment · Reviewer_fUf1 · 2021-09-01
> > > > **Paper must be refocused on stack RNNs**
> > > >
> > > > Thank you for the quick response. From this comment and from your initial response, it appears that any (discrete) stack RNN can be trivially rewritten as an RNN with growing memory module (by direct transfer of weights and using your description in the first response), and vice versa (by direct transfer of weights and using your description in the second response). Ie, stack RNNs and growing memory modules are not merely equivalent but isomorphic (as you write above).
> > > >
> > > > Hence, this paper should be rewritten throughout to be about stack RNNs. I.e, instead of ”proposing” the growing memory module (as is done in the current draft), it should simply state that it will be about stack RNNs, present the description of them using the familiar descriptions in the literature, and then update the proofs so they are presented on stack RNNs. (This is because it would be confusing (and detrimental to the field) to bloat it with new models and names when they are in fact isomorphic to existing ones.)

---

> > > > > ### Author Response · Authors · 2021-09-01
> > > > > **Response to R1**
> > > > >
> > > > > We thank the reviewer for the suggestion. There is an important misunderstanding that we would like to clarify. To aid discussion, we use the following definition: Two systems, X and Y, are *isomorphic* if X can simulate Y and Y can simulate X. Two systems, X and Y, are *equivalent* if X and Y can be trivially re-written as one another.
> > > > >
> > > > > Our proposed growing memory modules are isomorphic to most proposed stack RNNs but *not* equivalent to them. Our initial response is on how a stack RNN can simulate a growing memory module (it seems that the reviewer misunderstood it as the opposite direction). We repeated it in the proof of (i) in our response today. As for the opposite direction, the formulas of most stack RNNs cannot be re-written as the formulas in the growing memory module due to the differentiable mechanism (this is why we rely on the Turing completeness to establish the proof of (ii)). The growing memory module is designed as a foundation form of stack RNNs, so different forms of stack RNNs can simulate it easily.
> > > > >
> > > > > Moreover, "stack RNN" in general refers to a class of RNNs that employ a stack-like mechanism, and there is no universal definition of the term. Many forms of stack RNNs, such as Neural Stack [1], Neural Network Pushdown Automata (NNPDA) [2], and DiffStk-RNN [3], have been proposed, with each having different terminology and symbols. Though they are all isomorphic to one another (they are all Turing-complete, as shown by our paper), they are not equivalent. Each of these stack RNNs holds significance (mostly empirical performance) on its own. Similarly, our proposed growing memory module is not equivalent to any proposed stack RNNs and holds theoretical significance on its own. The growing memory module is explicitly designed to allow easy extension of proof to other stack RNNs and bounded-precision RNNs, as shown in Theorem 6. Therefore, the term "propose" is appropriate given the novelty of the design. However, we acknowledge that we should discuss the relation between stack RNNs and the proposed growing memory module thoroughly in the revised draft. Most importantly, we will emphasize that *the proposed growing memory module belongs to the class of stack RNNs and can be easily simulated by other proposed stack RNNs*.
> > > > >
> > > > > (If we limit our proof to a particular type of stack RNN such as [1], then the proof cannot be easily generalized to other stack RNNs such as [2], [3], and bounded-precision RNNs. The design of the growing memory module allows easy simulation by other stack RNNs and bounded-precision RNNs, so the theorems of it can be easily generalized to other RNNs as shown in Theorem 6.)
> > > > >
> > > > > [1] Grefenstette, E., Hermann, K. M., Suleyman, M., & Blunsom, P. (2015). Learning to transduce with unbounded memory. Advances in neural information processing systems, 28, 1828-1836.
> > > > >
> > > > > [2] Sun, G. Z., Giles, C. L., Chen, H. H., & Lee, Y. C. (2017). The neural network pushdown automaton: Model, stack and learning simulations. arXiv preprint arXiv:1711.05738.
> > > > >
> > > > > [3] Mali, A., Ororbia, A., Kifer, D., & Giles, C. L. (2020). Recognizing long grammatical sequences using recurrent networks augmented with an external differentiable stack. arXiv preprint arXiv:2004.07623.

---

> > > > > ### Author Response · Authors · 2021-09-01
> > > > > **Additonal Section**
> > > > >
> > > > > Given the extended discussion with the reviewer on the relationship of the growing memory module with the stack RNNs, which we acknowledge to be important, we will add a separate section in the revised draft as below.
> > > > >
> > > > > At last, we would also like to emphasize that the contribution of our paper lies in the proof of the Turing completeness of (i) unbounded-precision RNNs, (ii) bounded-precision stack-augmented RNNs, (iii) bounded-precision RNNs. The proof on (iii) is arguably the most important contribution of the paper and we hope that the reviewer can recognize its significance despite the confusion in (ii).
> > > > >
> > > > > **4.1 Relationship of the Growing Memory Modules with Stack-augmented RNNs**
> > > > >
> > > > > The proposed growing memory module belongs to the generic class of stack-augmented RNNs, which refers to any RNNs that are augmented with a stack-like mechanism. Many different forms of stack-augmented RNNs have been proposed, such as neural stack [1], stack RNN [2], NNPDA [3], NSPDA[4], DiffStk-RNN [5]. Given the simplicity of the design, the growing memory module represents the foundation form of these stack-augmented RNNs - it is easy to show that these stack-augmented RNNs can simulate the growing memory module easily in linear time. For example, the growing memory modules can be simulated by the neural stack [1] as follows. For pushing operation, set $v_t$ in the neural stack to $z_t$ in the growing memory module, and $d_t$ in the neural stack to $1[z_t > 0]$ in the growing memory module. For popping operation, set $u_t$ in the
> > > > > neural stack to $1[k_t = 0]$ in the growing memory module. Therefore, it follows that the proof of Turing completeness of bounded-precision RNNs with growing memory modules can extend to other stack-augmented RNNs. That is, bounded-precision stack-augmented RNNs [1-5] are also Turing-complete. Thus, Theorem 4 also provides theoretical motivation for stack-augmented RNNs in general.
> > > > >
> > > > > Different from other stack-augmented RNNs, the proposed growing memory modules use a simple mechanism to control pushing and popping - only the top neurons in the stack are included in the RNNs. This allows theories relating to the proposed growing memory modules to be easily extended to other forms of RNNs, including bounded-precision RNNs (without any stack), as shown in the following section (Section 5).
> > > > >
> > > > > [1] Grefenstette, E., Hermann, K. M., Suleyman, M., & Blunsom, P. (2015). Learning to transduce with unbounded memory. Advances in neural information processing systems, 28, 1828-1836.
> > > > >
> > > > > [2] Joulin, A., & Mikolov, T. (2015). Inferring algorithmic patterns with stack-augmented recurrent nets. Advances in neural information processing systems, 28, 190-198.
> > > > >
> > > > > [3] Sun, G. Z., Giles, C. L., Chen, H. H., & Lee, Y. C. (2017). The neural network pushdown automaton: Model, stack and learning simulations. arXiv preprint arXiv:1711.05738.
> > > > >
> > > > > [4] Mali, A. A., Ororbia II, A. G., & Giles, C. L. (2020). A Neural State Pushdown Automata. IEEE Transactions on Artificial Intelligence, 1(3), 193-205.
> > > > >
> > > > > [5] Mali, A., Ororbia, A., Kifer, D., & Giles, C. L. (2020). Recognizing long grammatical sequences using recurrent networks augmented with an external differentiable stack. arXiv preprint arXiv:2004.07623.

---

### Decision · Program_Chairs · 2021-09-27

**Decision:**

Accept (Poster)

**Comment:**

This paper has gone through deep discussions among the reviewers. Assuming all the proofs are correct, a major weakness of this paper is its disconnection with the community. The stack-RNNs and related models are not discussed in the paper. It is unclear what the community can learn from this work. However, the contribution that fixed precision RNN models with stacks are as powerful as a Turing machine is novel and significant. The analyzed RNN model is more or less practical. The theory may have a wide impact. Meanwhile, Meanwhile, the authors are highly encouraged to complete the connection to existing related work.